# Repertoire characterization and validation of gB-specific human IgGs directly cloned from humanized mice vaccinated with dendritic cells and protected against HCMV

Sebastian J. Theobald[1,2,3‡], Christoph Kreer[4,5‡], Sahamoddin Khailaie[6], Agnes Bonifacius[7], Britta Eiz-Vesper[7], Constanca Figueiredo[7], Michael Mach[8], Marija Backovic[9], Matthias Ballmaier[10], Johannes Koenig[1,2,3], Henning Olbrich[1,2,3], Andreas Schneider[1,2,3], Valery Volk[1,2,3], Simon Danisch[1,2,3], Lutz Gieselmann[4,5,11], Meryem Seda Ercanoglu[4,5], Martin Messerle[3,12], Constantin von Kaisenberg[13], Torsten Witte[14], Frank Klawonn[15,16], Michael Meyer-Hermann[6,17,18], Florian Klein[4,5,11], Renata Stripecke[1,2,3]*

1 Clinic of Hematology, Hemostasis, Oncology and Stem Cell Transplantation, Hannover Medical School, Hannover, Germany, 2 Laboratory of Regenerative Immune Therapies Applied, Excellence Cluster REBIRTH, Hannover Medical School, Hannover, Germany, 3 German Centre for Infection Research (DZIF), Partner Site Hannover-Braunschweig, Hannover, Germany, 4 Laboratory of Experimental Immunology, Institute of Virology, Faculty of Medicine and University Hospital of Cologne, University of Cologne, Cologne, Germany, 5 Center for Molecular Medicine Cologne (CMMC), University Hospital of Cologne, Cologne, Germany, 6 Department of Systems Immunology and Braunschweig Integrated Centre of Systems Biology (BRICS), Helmholtz Centre for Infection Research, Braunschweig, Germany, 7 Institute of Transfusion Medicine and Transplant Engineering, Hannover Medical School, Hannover, Germany, 8 Institute of Virology, University Erlangen-Nürnberg, Erlangen, Germany, 9 Structural Virology Unit, Department of Virology, Institut Pasteur, Paris, France; CNRS UMR 3569, Paris, France, 10 Research Facility Cell Sorting, Hannover Medical School, Hannover, Germany, 11 German Centre for Infection Research, Partner Site Bonn-Cologne, Cologne, Germany, 12 Instiute of Virology, Hannover Medical School, Hannover, Germany, 13 Department of Obstetrics, Clinic of Gynecology and Reproductive Medicine, and Obstetrics, Hannover Medical School, Hannover, Germany, 14 Department of Rheumatology and Immunology, Hannover Medical School, Hannover, Germany, 15 Biostatistics Group, Helmholtz Centre for Infection Research, Braunschweig, Germany, 16 Institute for Information Engineering, Ostfalia University, Wolfenbuettel, Germany, 17 Institute for Biochemistry, Biotechnology and Bioinformatics, Technische Universität Braunschweig, Braunschweig, Germany, 18 Cluster of Excellence RESIST (EXC 2155), Hannover Medical School, Hannover, Germany

‡ These authors share first authorship on this work.
* stripecke.renata@mh-hannover.de

## Abstract

Human cytomegalovirus (HCMV) causes serious complications to immune compromised hosts. Dendritic cells (iDCgB) expressing granulocyte-macrophage colony-stimulating factor, interferon-alpha and HCMV-gB were developed to promote *de novo* antiviral adaptive responses. Mice reconstituted with a human immune system (HIS) were immunized with iDCgB and challenged with HCMV, resulting into 93% protection. Immunization stimulated the expansion of functional effector memory CD8$^+$ and CD4$^+$ T cells recognizing gB. Machine learning analyses confirmed bone marrow T/CD4$^+$, liver B/IgA$^+$ and spleen B/IgG$^+$ cells as predictive biomarkers of immunization ($\approx$87% accuracy). CD8$^+$ and CD4$^+$ T cell responses against gB were validated. Splenic gB-binding IgM$^-$/IgG$^+$ B cells were sorted and analyzed at a single cell level. iDCgB immunizations elicited human-like IgG responses with

**Data Availability Statement:** All relevant data are within the manuscript and its Supporting Information files.

**Funding:** R.S./S.T./V.V./H.O. Hannover: This work was financed by grants of the German Center for Infections Research (DZIF-TTU07.803 and DZIF-TTU07.805 to R.S.), by a research collaboration grant of "The Jackson Laboratory" and by the German Research Council (DFG/SFB738 Project A6 to R.S. and MM; DFG/REBIRTH Unit 6.4 to R.S.). S.T. received a RegSci Ph.D. fellowship, H.O. received a DZIF-Strucmed fellowship and V.V. received a DAAD/ZIB Ph.D. fellowship. F.K./C.K./ Univ. Cologne: This work was funded by grants from the German Center for Infection Research (DZIF, F.K.), the German Research Foundation (CRC 1279, F.K.; CRC 1310, C.K. and F.K.; Heisenberg-Program KL2389/2-1, F.K.) and the European Research Council (ERC-StG639961, F. K.). M.M.-H./ S.K./ Braunschweig: S.K. was supported by the German Federal Ministry of Education and Research (BMBF) for the eMED project SYSIMIT and by the Helmholtz-Gemeinschaft, Zukunftsthema "Immunology and Inflammation" (ZT-0027). The funders had no role in study design, data collection and analysis, decision to publish, or preparation of the manuscript.

**Competing interests:** I have read the journal's policy and the authors of this manuscript have the following competing interests: the corresponding author is a co-inventor in a patent related to the content of the manuscript: R. Stripecke et al, "Induced dendritic cells and uses thereof" US patent No: US10,272,111 B2. RS received honoraria and funding support from The Jackson Laboratory.

a broad usage of various IgG heavy chain V gene segments harboring variable levels of somatic hypermutation. From this search, two gB-binding human monoclonal IgGs were generated that neutralized HCMV infection *in vitro*. Passive immunization with these antibodies provided proof-of-concept evidence of protection against HCMV infection. This HIS/ HCMV *in vivo* model system supported the validation of novel active and passive immune therapies for future clinical translation.

## Author summary

Human cytomegalovirus (HCMV) is a ubiquitous pathogen. As long as the immune system is functional, T and B cells can control HCMV. Yet, for patients who have debilitated immune functions, HCMV infections and reactivations cause major complications. Vaccines or antibodies to prevent or treat HCMV are not yet approved. Novel animal models for testing new immunization approaches are emerging and are important tools to identify biomedical products with a reasonable chance to work in patients. Here, we used a model based on mice transplanted with human immune cells and infected with a traceable HCMV. We tested a cell vaccine (iDCgB) carrying gB, a potent HCMV antigen. The model showed that iDCgB halted the HCMV infection in more than 90% of the mice. We found that antibodies were key players mediating protection. Using state-of-the-art methods, we were able to use the sequences of the human antibodies generated in the mice to construct and produce monoclonal antibodies in the laboratory. Proof-of-concept experiments indicated that administration of these monoclonal antibodies into mice protected them against HCMV infection. In summary, this humanized mouse model was useful to test a vaccine and to generate and test novel antibodies that can be further developed for human use.

## Introduction

Human cytomegalovirus (HCMV) is a broadly spread herpes virus, with an estimated sero-prevalence of 83% in the general population worldwide [1]. The course of HCMV infection largely depends on the host immune status [2]. Congenital or neonatal HCMV infection remains as an unmet clinical need affecting newborns and is associated with serious sequelae such as mental retardation or hearing loss [3]. In immune-compromised patients, HCMV reactivation from latency can cause high morbidity and mortality, for instance in patients undergoing hematopoietic stem cell transplantation (HCT), solid organ transplantation (SOT) or carrying a human immunodeficiency virus (HIV) infection [4]. Early reactivation after HCT is associated with low overall survival and clinical manifestations like pneumonia, colitis or retinitis [4, 5]. HCMV remains still a major clinical problem, despite the existence of antiviral drugs and the recent success of letermovir in clinical trials [4, 6–8]. Therefore, the development of vaccines or passive immunization with monoclonal antibodies such as hyperimmunoglobulin-preparations (HIG) containing high concentrations of HCMV-specific IgGs are of particular importance [9]. Several clinical trials targeting different HCMV antigens, such as glycoprotein B (gB), the pentameric complex (PC) or phosphoprotein 65 (pp65), have been conducted [9, 10]. However, none of them induced full protection and up to date there is no vaccine or monoclonal antibody approved for clinical use.

One factor that has considerably delayed the translation of novel passive or active immunization approaches to clinical trials has been the lack of practical and reproducible animal models relevant to HCMV research [11]. Humanized mice transplanted with CD34$^+$ hematopoietic stem cells (HSCs) and reconstituting a human immune system (HIS) have been broadly used to model human infection diseases [12] and to study human innate and adaptive immune responses *in vivo* [13–15]. Some groups have developed humanized mouse models using different immune deficient mouse strains implanted with cord blood (CB) hematopoietic cells or fetal tissues to recapitulate HCMV infection and reactivation [11]. We recently showed that *NOD.Cg-Rag1*$^{tm1Mom}$*IL-2Rγ*$^{tm1Wjlc}$ (NRG) mice transplanted with CB HSCs acquiring long-term human immune reconstitution could be reproducibly challenged with a traceable HCMV strain expressing the Gaussia Luciferase (GLuc) and the infection was traceable by non-invasive optical imaging analyses [16]. In line with the model described by Nelson and Caposio [17], in our humanized mouse model the infection leads primarily to viral latency. In these HCMV-infected mice, reactivation could be induced seven weeks later by daily administrations of Granulocyte-Colony Stimulating Factor (G-CSF). This "reactivation", analyzed a week after G-CSF treatment was initiated, produced a significant elevation of the optical imaging values analyzed eight weeks later relative to infection without G-CSF treatment [16]. Detection of HCMV viral copies by PCR showed that HCMV was detectable after infection in CD34$^+$ HSCs and CD14$^+$ monocytes, whereas after G-CSF treatment and reactivation the viral copies were conspicuously more abundant in CD34$^+$ HSCs and CD169$^+$ macrophages [16]. Although no specific viral latency marker was evaluated, the infection without final G-CSF treatment showed characteristics of viral latency. Infections were further correlated with development of memory CD4$^+$ T cells, whereas reactivation induced an increase of PD-1$^+$ activated/ exhausted T cells [16]. This model also showed that HCMV reactivation was associated with significant maturation of human IgG$^+$ plasma B cells [16]. Other noteworthy HCMV infection models have been reported [11], but the advantage of our model is the relatively simple non-invasive monitoring of the HCMV infection and the possibility of highly sensitive amplification of the viral detection after G-CSF treatment in order to predict if a vaccine or therapy can control or mitigate a primary viral infection. Thus, this model is useful for testing the protective effects of human-specific vaccines or immunotherapies and to evaluate the distinct immunological mechanisms that block or reduce a HCMV primary infection.

Dendritic cells (DCs) express high levels of HLA, costimulatory ligands and cytokines, and play a pivotal role as antigen-presenting cells in the priming and boosting of naïve T and B cells. Previously, we have supplemented HIS mice with "induced DCs" (iDCs) that can self-differentiate *in vivo* and stimulate immune responses because they expressed human granulocyte-macrophage colony-stimulating factor (GM-CSF), interferon-alpha (IFN-α), and the HCMV phosphoprotein (pp)65 antigen. iDCpp65 stimulated potent and long-lasting T cell responses in humanized NRG mice [18–20].

The envelope glycoprotein B (gB) of HCMV is a major target and highly studied antigen for generation of virus neutralizing antibodies [21]. We therefore hypothesized that expression of the full-length gB in its native conformation on the surface of iDCgB would be efficient to promote humoral responses to halt HCMV infection and for long-lasting control of HCMV reactivation. Here, we demonstrate that iDCgB mediated potent, long-lasting and specific immune protection against HCMV primary infections in HIS mice. This protection was associated with improved development of mature human T and B cells and specific immune responses against gB with no side effects. Immunization of HIS mice with iDCgB generated high frequencies of gB-binding IgG$^+$ cells, enabling efficient direct discovery of fully human gB-binding monoclonal antibodies (mAbs). These antibodies neutralized HCMV *in vitro* and protected HIS mice against HCMV primary infections *in vivo*. Therefore, we describe how this humanized mouse

model was useful to validate a novel vaccine *in vivo* and to discover fully human monoclonal antibodies neutralizing HCMV.

## Results

### iDCgB immunization and T cell responses

We explored the HCMV-gB protein as an antigen for immune protection of humanized mice against HCMV. For production of iDCgB, CD14$^+$ monocytes were transduced for 16 h with lentiviral vectors (LVs: LV-GMCSF-2A-IFNα+LV-gB/ or LV-GMCSF-2A-IFNα-2A-gB) and cryopreserved. After thawing and *in vitro* culture for seven days, viable cells expressed HLA-DR, costimulatory markers (CD80 and CD86) and surface expression of gB (**Fig 1A**).

*Nod.Rag.Gamma* (NRG) mice generated with three different cord blood (CB) donors were humanized with CD34$^+$ cells. One cohort ("iDCgB", n = 14) was immunized with same donor iDCgB and the other was not treated ("CTR", n = 8). Long-term memory responses were analyzed at 14 weeks after the last immunization (i.e., 25 weeks after HCT; **Fig 1B**). Longitudinal human CD45 reconstitution in peripheral blood (huCD45$^+$ in BL) was comparable between the control and iDCgB-immunized mice (50%), but the frequencies of human CD3$^+$ T cells and numbers of detectable lymph nodes (LN) at various anatomical sides were elevated for the iDCgB group (**Fig 1C and 1D**). This was associated with increased concentrations of several human cytokines detectable in plasma with a mixed T-helper (Th) 1/2 pattern (significantly higher values were seen for GM-CSF, IL-10, IL-12; higher values for IL-5, IFN-γ, MCP-1 and TNF-α; **Fig 1E**). Analyses of CD4$^+$ and CD8$^+$ T cell subtypes present in different lymphatic tissues were performed (representative gating example for T cell analysis is shown in **S1A Fig**). The total numbers of CD4$^+$ and CD8$^+$ central memory (CM), effector memory (EM) and terminal effector (TE) T cells were significantly increased in mesenteric lymph nodes (mLN) of the iDCgB group compared with CTR (**Fig 1F**). Analyses of peripheral lymph nodes (LN), bone marrow (BM) and spleen (SPL) also revealed enhanced expansion of memory and effector T cell subsets upon iDCgB immunization (**S1, S1D; S1E Fig; S1 Table**). Analyses of the relative frequencies of T cell subtypes in different tissues pointed to a consistent expansion of EM CD4$^+$ and CD8$^+$ T cells after iDCgB immunization (**S1F Fig; S1 Table**). The total numbers and frequencies of follicular Th cells (fTh: CD45$^+$/CD4$^+$/PD-1$^+$/CXCR5$^+$; gating strategy shown in **S2E Fig**) detected in LN were higher in the iDCgB cohort (grey, n = 14) compared with CTR (green, n = 7). The functionality of cryopreserved cells recovered from mesenteric LN was evaluated *in vitro* (CTR: 2 pooled mLN samples; iDCgB: 3 pooled mLN samples). The cells were thawed, cultured with huIL-7 and huIL-15 for 4 h and then they either received no antigen stimulation or were pulsed with gB or pp65 recombinant proteins for 1 h. Later, the cells were further incubated with huIL-7 and huIL-15 and with Brefeldin A to block the secretory pathway for 15 h (scheme shown in **S2A Fig**). After the *in vitro* culture, T cells were analyzed by flow cytometry. EM CD45$^+$CD4$^+$ or CD45$^+$CD8$^+$ T cells were analyzed for detection of intracellular IFN-γ and TNF-α by intracellular staining (ICS) (gating strategy is shown in **S2B Fig**). T cells recovered from iDCgB-immunized mice and pulsed with gB protein showed clearly detectable IFN-γ$^+$ and TNF-α$^+$ CD8$^+$ (top panels) and CD4$^+$ (lower panels) EM T cells (**Fig 1H**). Only background levels of IFN-γ$^+$ and TNF-α$^+$ cells were detectable from cells non-pulsed or pulsed with an irrelevant protein (pp65) of from cells recovered from CTR mice (**Fig 1H;** additional experimental arm using T cells from a HCMV seropositive human donor and non-specific stimulation with PMA-Ionomycin are shown in **S2C Fig**). To conclude, iDCgB immunization enhanced the levels of human cytokines in plasma, expansion of memory CD4$^+$ and CD8$^+$ T cells, follicular T helper cells, and functional gB-specific CD4$^+$ and CD8$^+$ T cell responses.

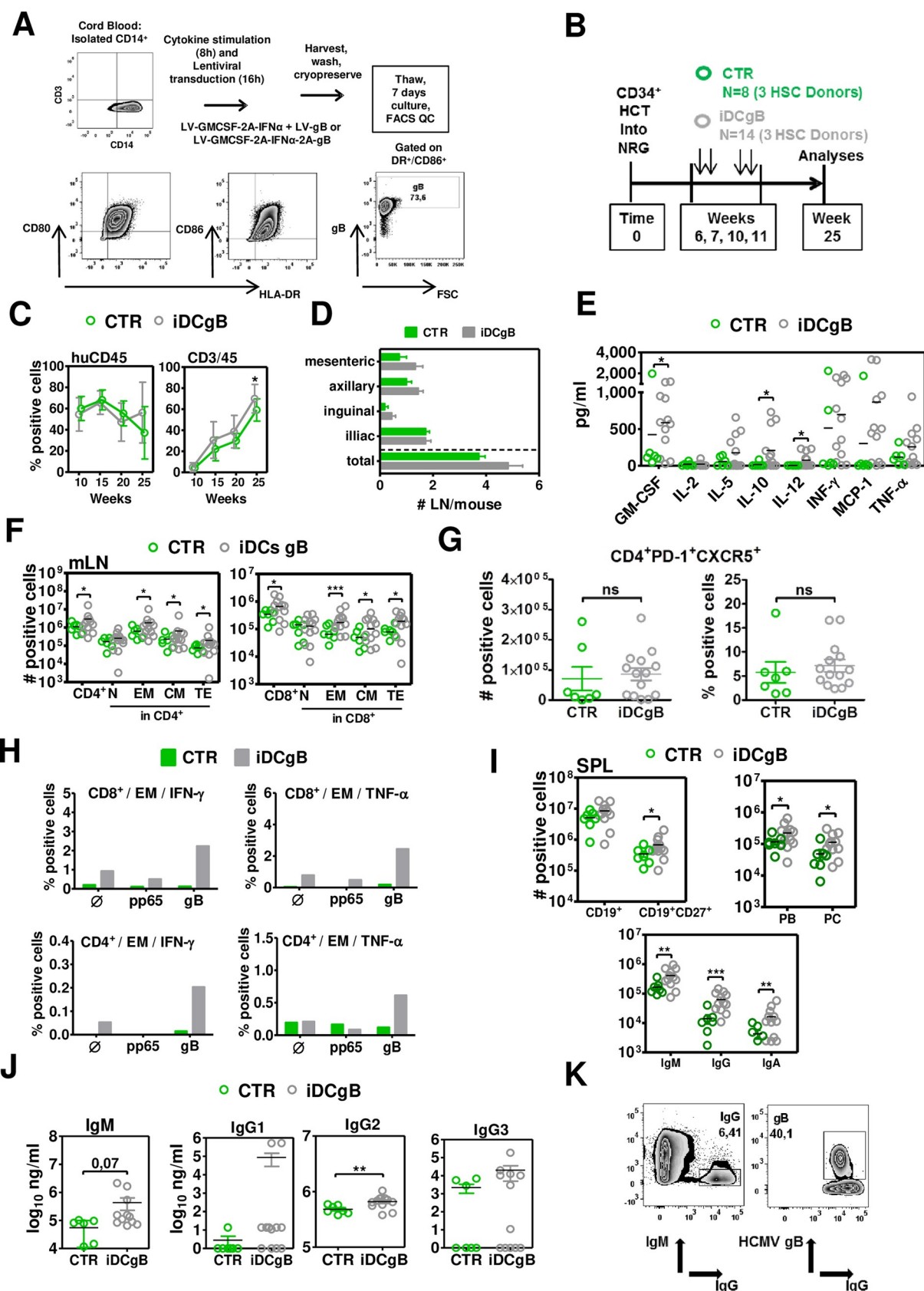

**Fig 1. Post-transplant immunization of humanized NRG mice with iDCgB improved the maturation of T and B cells and enhanced the humoral responses. A)** Scheme of iDCgB generation. Lentiviral vectors (LV-GMCSF-2A-IFNα + LV-gB or LV-GMCSF-2A-IFNα-2A-gB) were used to transduce CB-CD14$^+$ monocytes and after 16 h the cells were washed and cryopreserved. After thawing, the cells were maintained in culture for seven days. Right panels: Analyses of iDCgB (in this case two LVs were used for co-transduction) by flow cytometry analyses confirming the DC identity (HLA-DR$^+$/CD80$^+$, HLA-DR$^+$/CD86$^+$) and expression of gB on the cell surface. A representative example is shown from more than three experiments for iDCgB generated with two vectors or one tricistronic vector. **B)** Scheme of HCT into NRG mice using CB-CD34$^+$ stem cells and immunizations with CB-matched iDCgB. Highly purified CD34$^+$ cells obtained from three CB donors were used to transplant 5–6 weeks old female NRG mice. At 6, 7, 10 and 11 weeks after HCT, mice were immunized s.c. with 5x10$^5$ iDCgB (n = 14; grey color code) or were not treated (CTR; n = 8; green color code). Blood draws and analyses were performed at weeks 10, 15 and 20 after HCT and euthanasia was performed at week 25. **C)** Kinetics of human hematopoiesis in blood. Left panel shows comparable long-term frequencies of human CD45 in mice of CTR (green) and iDCgB (grey) cohorts. Right panel shows higher frequencies of human CD3$^+$ T cells in iDCgB (grey) cohort. **D)** Mesenteric, axillary, inguinal, iliac and total detectable lymph nodes were quantified. Graph shows the averages with a standard deviation of the mean for CTR (green) and iDCgB (grey) cohorts. **E)** Human cytokines (pg/ml) detectable in plasma. Each dot represents one mouse (CTR n = 6 samples available, green; iDCgB n = 12 samples available, grey). **F)** Total counts (# positive cells) for huCD45$^+$/CD4$^+$ T cells (left) and huCD45$^+$/CD8$^+$ T cells (right) measured for different subtypes (N, CM, EM, TE) in CTR (green dots) or iDCgB cohort (grey dots) in mLN. Representative gating example for CD45RA and CD62L staining is shown in S1 Fig. Each dot represents one mouse. **G)** Left panels: Total numbers of follicular T helper cells detected in LN (fTh: CD45$^+$/CD4$^+$/PD-1$^+$/CXCR5$^+$) of CTR (green, n = 7) and iDCgB (grey, n = 14) cohorts. Right panel: Percentage of fTh cells (% positive cells) for the same cohorts. Gating strategy is shown in S2 Fig. **H)** *In vitro* analyses of T cell reactivity against gB. mLN of the mice for each cohort were pooled, cohorts were analyzed separately and then the data was merged (HCMV: blue; n = 2 cohorts; iDCgB /HCMV:red; n = 3 cohorts). The cells were supplemented with cytokines, stimulated with recombinant proteins (pp65 or gB) *in vitro* and analyzed by flow cytometry (Ø denotes no antigen stimulation). EM CD8$^+$ or CD4$^+$ cells were gated and IFN-γ$^+$ or TNF-α$^+$ cells were quantified. The schematic representation of the assay and exemplary gating approaches are shown in S1 Fig. Left panels: Detection of IFN-γ expression in CD8$^+$ (top) or CD4$^+$ (lower panel) T cells. Left panels: Detection of TNF-α expression in CD8$^+$ (top) or CD4$^+$ (lower panel) T cells. **I)** Total counts (# positive cells) of pan-B cells (CD19$^+$), memory B cells (CD19$^+$CD27$^+$), IgA$^+$/ IgG$^+$/ IgM$^+$ B cells, plasmablasts (PB) and plasma cells (PC) measured in CTR (green dots) or iDCgB cohort (grey dots). (Exemplary gating approach is shown in S1 Fig). **J)** Total human IgM (μg/ml) and IgGs (types 1, 2, 3; μg/ml) measured in plasma by bead-array for CTR (green dots) and iDCgB (grey dots) cohorts. **K)** Representative example showing detection of IgM$^-$IgG$^+$ positive B cells binding to gB recovered from SPL of iDCgB immunized mice. Left panel: Gating of IgM$^-$/IgG$^+$ cells. Right panel: Gating of gB$^+$ binding cells showing frequency within the IgM$^-$/ IgG$^+$ population. Exemplary gating approach with all markers used is shown in Supplementary Material. The experiment was performed as independent duplicates with similar results. For statistical analysis negative binomial (total cell counts) and beta regression (percentage) analysis were performed and significances are indicated *p<0.05, **p<0.01, ***p<0.001. For all other statistics students t-Test with Welch's corrections was used. Standard deviation of the mean is indicated.

## iDCgB immunization and B cell responses

The next step was to evaluate the effects of iDCgB immunization on the B cell responses. Significant increases in the total numbers of CD19$^+$CD27$^+$ IgG$^+$, IgA$^+$, IgM$^+$ memory B cells and plasma cells (PC) were observed in SPL of the iDCgB group compared with controls (**Fig 1I**, **S1 Table** representative gating example in **S1B Fig**). Analyses of human antibody concentration in plasma showed higher levels of IgM (p = 0.07) and IgG$_2$ (p<0.01) in the iDCgB cohorts, and for some analyzed mice the concentrations of IgG$_1$ and IgG$_3$ were particularly increased (**Fig 1J**). B cells recovered from SPL were analyzed for their specific gB-binding capacity by flow cytometry (gating strategy shown in **S3 Fig**). 9–50% of the IgM$^-$IgG$^+$ B cells in iDCgB-immunized mice were reactive against gB (**Fig 1K**, for gating strategy see **S5 Fig**). In brief, iDCgB immunization promoted the reconstitution of IgG$^+$, IgA$^+$, IgM$^+$ memory B cells and development of gB-binding IgG$^+$ cells.

## iDCgB immunization against HCMV

Humanized mice engrafted with human hematopoietic cells from cord blood or fetal tissues are currently being used by several laboratories to study HCMV infection, reactivation and associated immune effects *in vivo* [11]. In previous work, our laboratory reported the establishment of a humanized mouse models for non-invasive dynamic tracking of a genetically modified HCMV strain expressing Gaussia luciferase (HCMV-GLuc) that recapitulated HCMV infection and reactivation [16]. Here, this model was used to evaluate primarily the prophylactic effects of iDCgB vaccination against a HCMV primary infection. If protection was not complete and a latent viral reservoir was eventually established, the virus could be reactivated with

a G-CSF stimulus to induce viral outburst to facilitate monitoring. Humanized mice were pre-immunized with iDCgB (n = 17; 3 different CB donors; immunizations were administered at weeks 6, 7, 10 and 11 after HCT). Non-immunized mice were included as controls (n = 12). Seventeen weeks after HCT, all mice were challenged with HCMV-GLuc to produce a primary infection (**Fig 2A**).

From week twenty four to twenty five after HCT, once viral latent infection was established, mice were then treated daily with G-CSF to stimulate HCMV/GLuc reactivation in order to amplify a putative residual viral infection in mice, making the monitoring more sensitive and traceable (**Fig 2A**). In order to minimize the handling of the mice (which could potentially interfere with the experimental outcome), the analyses of the immunological responses and viral load were performed only at the experimental endpoint, one week after G-CSF reactivation. Both immunized and non-immunized cohorts showed similar stable long-term huCD45 and huCD3 immune reconstitution in blood (**Fig 2B**), but iDCgB-vaccinated mice showed higher numbers of detectable LN structures, indicating a typical immunization effect (**Fig 2C**). Bioluminescence signal was conspicuous in anatomical regions of liver (LI) and salivary glands of all HCMV-infected control mice but barely noticeable for 13/17 iDCgB-immunized mice (**Fig 2D**). The highly sensitive monitoring of the full-body bioluminescence signal indicated that a residual HCMV infection was still measurable in the iDCgB-immunized mice but the signals were significantly decreased compared with control mice (**Fig 2E**). HCMV viral loads in mice were quantified by qPCR analyses of genomic viral copies in DNA extracted from LI and BM. Blood was not analyzed, because in previous work we observed that qPCR analyses of blood were not quantitatively reliable, with a high frequency of false-negative results [16]. Significantly decreased levels of HCMV DNA load were measured in 93% of iDCgB-immunized mice (**Fig 2F**, **S2 Table**). HCMV was undetectable for 9/16 of the analyzed immunized mice for which DNA was available, in these cases indicating a full protection. These results showed that prophylactic iDCgB immunization efficiently minimized or even fully protected HIS mice against a primary HCMV infection. A residual HCMV load and spread could still be detectable in some of the vaccinated mice, with the caveat that G-CSF was purposely administered in order to amplify a putative residual latent viral reservoir and to make the readout of this model highly sensitive. Whether the prophylactic iDCgB vaccine could promote immunologic effects to mitigate reactivation of the latent reservoir *per se* remains to be evaluated with a different experimental design, i.e. with cross-sectional or longitudinal analysis.

## iDCgB immunization prior to HCMV infection and T cell responses

As iDCgB immunization resulted in lower levels of HCMV reactivation as the final experimental read-out, we examined if this outcome was also associated with long-term immunologic differences between immunized and non-immunized mice. Analyses of total numbers of cells recovered from SPL and BM showed approximately a tenfold increase for CD4+ and CD8+ T cells after iDCgB immunization compared with infected controls (**Fig 2G**). All CD8+ T cell subtypes were expanded in the iDCgB-immunized group, with a strong bias towards the effector cells. This trend was also observed for T cells recovered from LI, LN and mLN (**S3A–S3C Fig**, **S4–S7 Tables**). PD-1 is considered a marker of T cell activation and exhaustion and in previous work we showed that PD-1 expression was upregulated in humanized mice with HCMV reactivation [16]. Parenthetically, the frequencies of PD-1+ CD3+ T cells were significantly higher in LN of HCMV controls, whereas iDCgB immunization seemed to confer a shielding effect to this persistent activation pattern promoted by HCMV reactivation (**Fig 2H**, **S3D–S3F Fig**, **S4–S7 Tables**). We therefore investigated if iDCgB immunization followed by HCMV reactivation promoted a better performance in terms of HCMV-specific T cell

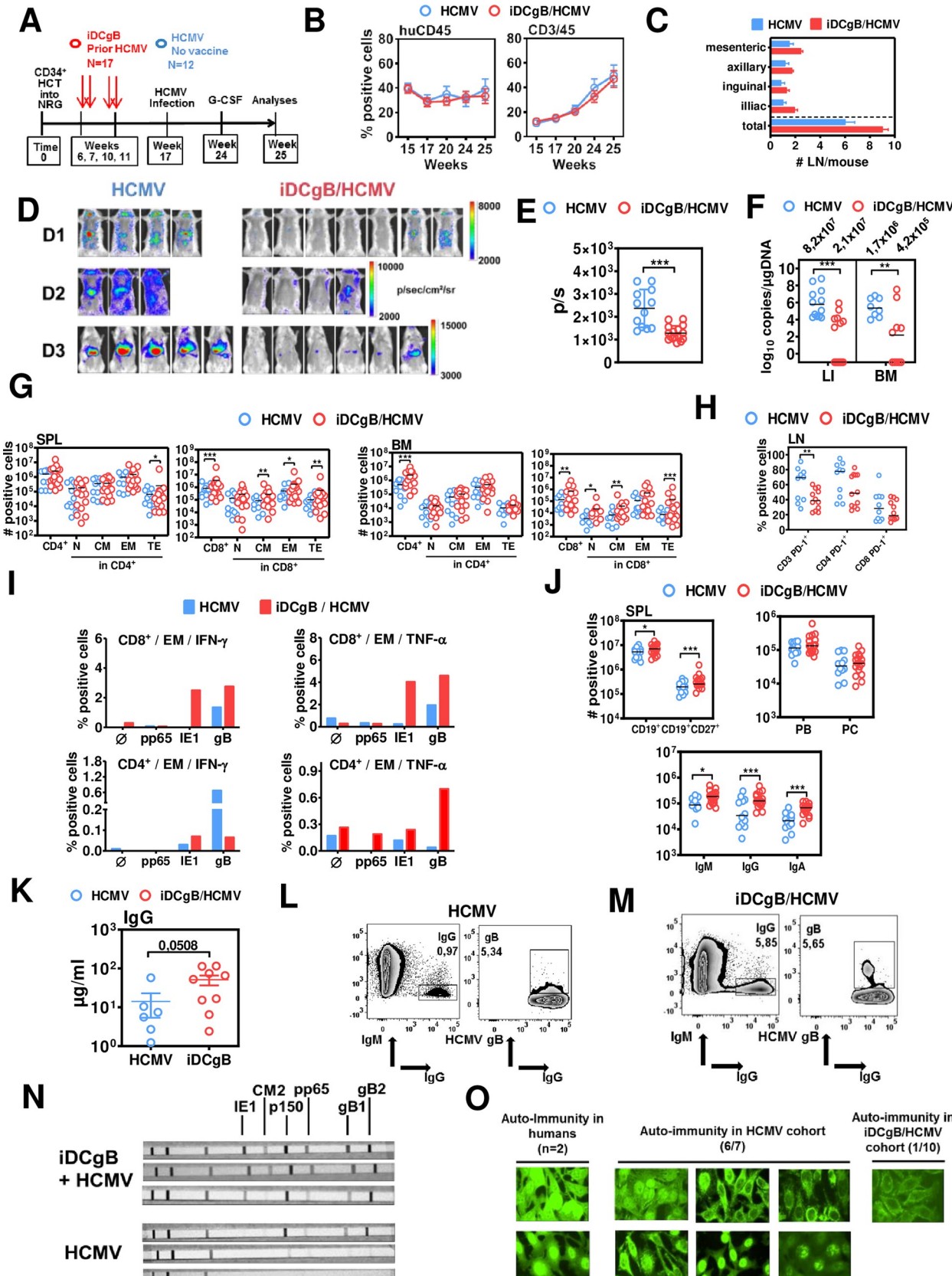

**Fig 2. Immunization of humanized mice with iDCgB protected against HCMV and promoted the persistence of long-term gB-specific functional memory CD8+ T cells and IgG+ B cells. A)** Experimental scheme of iDCgB immunization and HCMV infection in humanized mice.

Mice were transplanted with CB-CD34$^+$ cells and immunized with autologous iDCgB at weeks 6, 7, 10, 11 after HCT. HCMV infection was performed at week 17 post HCT with 1x10$^6$ HCMV/GLuc-infected MRC-5 cells (i.p.). From week 24 to 25, all mice were treated with granulocyte-colony stimulating factor (G-CSF, 2.5 μg/mouse/day) for 7 days in order to reactivate HCMV. **B)** Kinetics of human hematopoiesis in blood. The frequencies of human CD45$^+$ (left) and CD3$^+$ positive cells (right) were analyzed (at weeks 15, 17, 20, 24, 25) in blood of humanized mice comparing HCMV (blue) and iDCgB/HCMV cohorts (red). Human hematopoietic reconstitution was persistent and comparable for both cohorts. **C)** Mesenteric, axillary, inguinal, iliac and total detectable lymph nodes were quantified. Graph shows the averages with the standard deviation of the mean for HCMV (blue) and iDCgB/HCMV (red) cohorts. **D)** Optical imaging analysis of the bioluminescence signal detected in the torso and abdomen and displayed in photons/sec/cm$^2$/sr for all mice (left panels: HCMV; right panels: iDCgB/HCMV). Data was obtained with three independent experiments using cord blood from three donors (D1, D2, D3). Signal intensity was measured with the same settings for all cohorts and mice with 300 sec exposure time, f-stop 1 and medium binning. **E)** Quantified bioluminescence signal displayed in photons/sec showing decrease of signal in iDCgB/HCMV cohort (red dots) compared with HCMV cohort (blue dots). ROI was set the same for all mice encompassing the whole mouse body. A significant protective effect caused by iDCgB immunization is noticeable. For statistical analysis t-Test with Welch's correction was used and significances are indicated with ***p<0.001. Standard deviation of the mean is indicated. **F)** HCMV DNA viral copies were determined by quantitative PCR in LI (HCMV n = 12; iDCgB/HCMV n = 16 available) and BM (HCMV n = 8 available; iDCgB/HCMV n = 12 available). Viral copies are indicated in log$_{10}$ copies/μgDNA. For statistical analysis, t-Test with Welch's correction was used and significances are indicated with **p<0.01 and ***p<0.001. The standard deviation of the mean is indicated. **G)** Total counts (# positive cells) for huCD45$^+$/CD4$^+$ T cells (left) and huCD45$^+$/CD8$^+$ T cells (right) measured for cells recovered from SPL (left panels) and BM (right panels) for different subtypes (N, CM, EM, TE) in the HCMV (blue dots) or the iDCgB/HCMV cohort (red dots). Representative gating example for CD45RA and CD62L staining is shown in S1 Fig. Each dot represents one mouse. **H)** Frequencies of PD-1$^+$ CD3$^+$/ CD4$^+$/ CD8$^+$ T cells (% positive cells) in LN of HCMV (blue dots) or iDCgB/HCMV cohort (red dots). **I)** *In vitro* analyses of antigen-specific T cell responses. mLN of the mice for each cohort were pooled, cohorts were analyzed separately and then the data was merged (HCMV: blue; n = 1 cohort; iDCgB/HCMV: red; n = 2 cohorts). The cells were supplemented with cytokines, stimulated with recombinant proteins (pp65, IE1 or gB) *in vitro* and analyzed by flow cytometry (Ø denotes no antigen stimulation). EM CD8$^+$ or CD4$^+$ cells were gated and IFN-γ$^+$ or TNF-α$^+$ cells were quantified. The schematic representation of the assay and exemplary gating approaches are shown in S2 Fig. Left panels: Detection of IFN-γ expression in CD8$^+$ (top) or CD4$^+$ (lower panel) T cells. Left panels: Detection of TNF-α expression in CD8$^+$ (top) or CD4$^+$ (lower panel) T cells. **J)** Total counts (# positive cells) of pan-B cells (CD19$^+$), memory B cells (CD19$^+$CD27$^+$), IgA$^+$/ IgG$^+$/ IgM$^+$ B cells, plasmablasts (PB) and plasma cells (PC) measured in HCMV (blue dots) or iDCgB/HCMV cohort (red dots). Exemplary gating approach is shown in S1 Fig. Each experiment was repeated at least twice with similar results. Negative binominal regression analyses were performed for all statistical determinations of the flow cytometry analysis for the total cells (# positive cells). Beta regression analysis was performed for frequencies (% positive cells). Significances are depicted with *p<0.05, **p<0.01, ***p<0.001. **K)** Total human IgG measured in plasma by ELISA (μg/ml) for HCMV (blue dots, n = 6 available) or iDCgB cohort (red dots, n = 9 available) (p = 0.0508 was determined with t-Test with Welch's correction). **L), M)** Representative examples showing detection of IgM$^-$IgG$^+$ positive B cells binding to gB recovered from SPL of HCMV or iDCgB/HCMV cohorts, respectively. Left panels: Gating of IgM$^-$/IgG$^+$ cells. Right panels: Gating of gB$^+$ binding cells showing frequency within the IgM$^-$/IgG$^+$ population. Exemplary gating approach with all markers used is shown in Supplementary Material. **N)** Plasma was used in strips immunoassays for the qualitative determination of IgG-reactivity against HCMV antigens (IE1, CM2, p150, pp65, gB1 and gB2). Three representative stripes are shown for iDCgB/HCMV (top, note broad cross-reactivity against multiple antigens or HCMV (bottom). Qualitative results for all the mice analyzed are shown in S2 Table. **O)** Antinuclear antibodies (ANAs) that bind to contents of the cell nucleus were investigated. HEp-2 cells were used as a substrate to detect the auto-antibodies by indirect immunofluorescence. Human sera obtained from two subjects with autoimmunity disorders were used as reference showing bright signals in nucleus and occasionally in cytoplasm (left panels). Similar analyses performed with plasma obtained from mice in the HCMV cohort showed 6 out of 7 cases ANA positivity (middle panels). One single mouse of the 10 cases analyzed in the iDCgB/HCMV cohort showed faint immunofluorescence signals in the cytoplasm.

responses after *in vitro* stimulation (**Fig 2I**). T cells recovered from mLN were stimulated *in vitro* with gB or with other HCMV proteins (IE1 and pp65). Mice immunized with iDCgB contained fair frequencies of EM CD8$^+$IFN-γ$^+$ (average 3%) and EM CD8$^+$TNF-α$^+$ (average 4%) T cells reactive against both gB and IE1, indicating a polyvalent T cell reactivity. Notably only very low CD8$^+$ T cell reactivity against the antigens was detectable in mice with HCMV reactivation. Lower frequencies of CD4$^+$IFN-α$^+$ (average 0.05%) and CD8$^+$IFN-γ$^+$ (average 0.7%) T cells reactive against gB were detectable in iDCgB-immunized mice. In iDCgB-immunized mice, modest frequencies of EM CD4$^+$ TNF-α$^+$ (average 0.7%) T cells reactive against gB were detectable after HCMV reactivation, while frequencies of gB-specific EM CD4$^+$IFN-γ$^+$ T cells were low in these mice (average 0.05%). Concluding, iDCgB immunization prior to HCMV challenge was associated with long-term functional and polyvalent T cell responses.

## iDCgB immunization prior to HCMV challenge, B cell responses and auto-immunity

iDCgB immunization markedly improved B cell maturation and class-switch, shown by a discernible increase in the total numbers of CD27$^+$ memory B cells, IgG$^+$/IgA$^+$/IgM$^+$ B cells, PB

and PC numbers compared with HCMV controls for analyses of SPL (**Fig 2J**) and several other tissues such as BM, LI, LN and mLN (**S2G–S2J Fig**, **S4–S7 Tables**). Human IgG concentration in plasma was elevated in immunized mice (**Fig 2K**), and 1–5% of the IgM⁻IgG⁺ B cells were able to bind to gB (**Fig 2L and 2M**, **S3 Fig**). Plasma was analyzed in a commercially available IgG immunoassay to evaluate humoral reactivity against several HCMV antigenic peptides blotted on strips (IE1, CM2, pp65, p150, and the subunits gB1 and gB2). Plasma from iDCgB/HCMV mice showed high reactivity against gB1 and gB2 and polyvalent responses to other antigens (most notably IE1 and p15), whereas this was not observed in non-immunized mice showing HCMV reactivation (**Fig 2N**, **S3 Table**). Additionally, plasma was tested regarding the presence of human antibodies for putative auto-immunity, i.e. reactive against human cells. For mice after HCMV reactivation, auto-reactive antibodies were observed in 6/7 tested mice. A faint reactivity was observed in plasma from only one out of 10 tested iDCgB-immunized mice (**Fig 2O**). In sum, iDCgB immunization prior to HCMV challenge promoted expansion of IgG⁺ memory B cells reactive against gB and IE1 and protected mice against humoral auto-immunity.

## Immunophenotypic signature of iDCgB immunization

The multifactorial analyses of several immunophenotypic markers from different tissues in several mice result into a myriad of very large datasets making the overall analyses and correlations very complicated. We previously employed machine learning (ML)-approaches for HIS mice to capture the most important multidimensional signatures of human adaptive immune reactivity[16, 19]. Here, feature selection and data classification identified subsets of biomarkers in tissues, discriminating between the HCMV and iDCgB/HCMV cohorts of mice. Initially, a Kolmogorov-Smirnov (KS) test was performed to explore the biomarkers that showed a difference between the groups. Due to the low sample size, a leave-one-out (LOO) cross-validation (CV) analysis was performed using datasets from SPL, BM and LI to identify the most reliable biomarkers that were insensitive to sample exclusion. Sample distribution of markers, which appeared 29-times during LOO-CV (**S4A Fig**), showed a clear distinction between both groups (**Fig 3A**).

The number of B cell associated markers was overall higher than the T cell associated ones. To obtain subsets of biomarker combinations that could best predict the sample group, a nested CV loop was performed (**S4A Fig**, as described in Material and Methods). The biomarkers with proper training and inner CV average classification accuracies summarized that IgA⁺, IgG⁺ and CD4⁺ T cells in SPL, LI and BM were predictive markers (**Fig 3B**). These markers were present in different sets to effectively discriminate between HCMV versus iDCgB/HCMV mice. The best set comprised of SPL total numbers of IgG⁺, LI frequency of IgA⁺ and BM total numbers of CD4⁺ with a classification accuracy of 87% (**Fig 3C**, **S8 Table**). A principal component analysis on the best classifying biomarkers sets confirmed that these measurements explained a large variation in the data obtained for iDCgB/HCMV and HCMV-challenged mice (**Fig 3D**). In conclusion, these results suggested that the total number and frequencies of IgA⁺ and IgG⁺ B cells were key predictors to define the iDCgB immunization effects prior to HCMV challenge.

## B cell depletion disturbs the protective effects against HCMV reactivation promoted by iDCgB

As the immunologic analyzes presented above indicated that iDCgB stimulated a predominant B cell humoral response, we questioned if B cell depletion would lower the immunologic control against HCMV. A clinical-grade anti-CD20 monoclonal antibody (rixathon, a rituximab

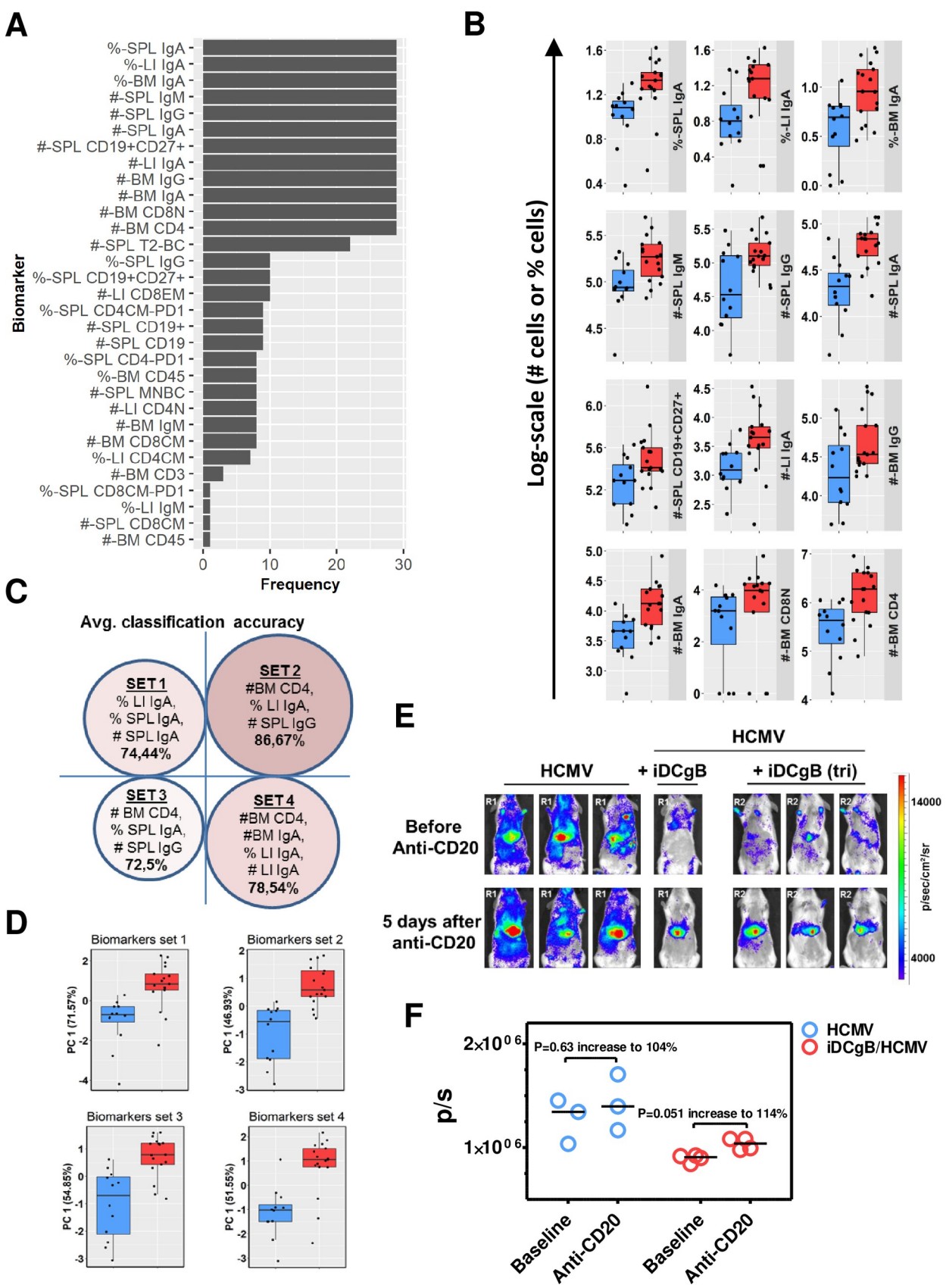

**Fig 3. A machine learning approach for data classification and identification of predictive biomarkers of immunization impact. A)** Kolmogorov-Smirnov (KS) test was performed on the cellular immunophenotypic biomarkers (without missing values) using all samples (incorporating both iDCgB/HCMV (n = 17); HCMV (n = 12) and in total n = 29). The sensitivity of the KS test on the sample size was evaluated by leave one-sample out (LOO), repeated for all samples. The thresholds for KS statistic and p-value were 0.3 and 0.1, respectively. The frequency of each biomarker satisfying the KS test criteria in LOO is shown. **B)** The sample distribution is shown for each single biomarker that satisfied the KS test thresholds and appeared with highest frequency in LOO. **C)** Four biomarker combination sets selected according to the pipeline (see Materials and Methods) and showed 73–87% accuracies to predict the iDCgB/HCMV group. **D)** Principal component analysis was performed for the four selected biomarkers sets (shown in c). The distribution of samples in the first principal component (PC1) is shown. **E)** Effects of anti-CD20 (B cell) depletion in the HCMV/GLuc infection signal. Top panels: Baseline optical imaging analyses of HCMV-infected mice (Reconstitution R1, HCMV, n = 3) or iDCgB immunized and then infected with HCMV (HCMV+iDCgB, n = 4; R1: iDCgB generated with 2 vectors; R2: iDCgB generated with a tricistronic lentiviral vector). Bottom panels: Optical imaging analyses 5 days after anti-CD20 (10 mg/kg) treatment for B cell depletion. All mice were analyzed with the same settings; the radiance of the bio-luminescence signals is indicated by the colored bar on the right side (p/sec/cm$^2$/sr). **F)** Quantified total Flux (photons/ second, p/s) for HCMV and HCMV+iDC/gB cohorts, before and after anti-CD20 depletion. ROI was quantified for the frontal torso and abdomen and kept constant for all mice. The horizontal bars in black indicate the median values for each cohort and time of analyses. The P values were determined by t-Test. The % increase is relative to the baseline values prior to depletion (see **S8** and **S9 Tables**).

biosimilar) was used in a pilot experiment to compare the effects of B cell depletion applied to HCMV-infected or to iDCgB-immunized and HCMV-infected mice. HCMV infection was monitored by optical imaging analyses at baseline before depletion and five days after B cell depletion (see experimental scheme and evidence of B cell depletion in **S4B–S4D Fig**; see core data and statistics in **S9 Table**). B cell depletion promoted only a mild increase in HCMV/ GLuc bioluminescence signal in non-immunized mice (n = 3; P = 0.63; 104% increase relative to baseline median values, **Fig 3E and 3F**, **S9A Table**). For the iDCgB-immunized and then HCMV-infected mice, the effects of B cell depletion were associated with a more conspicuous HCMV rebound (n = 4; P = 0.05; 114% increase relative to baseline median values, **Fig 3E and 3F**, **S9B Table**). Thus, these results indicated that the B cell responses elicited by iDCgB in this humanized model participated in the maintenance of HCMV immune control.

## Molecular analyses of B cell responses

Since a predominant B cell response was observed in iDCgB humanized mice, we investigated the anti-gB antibody response in more detail. To this end, single IgG$^+$ gB-specific B cells from SPL were sorted by flow cytometry in order to evaluate the anti-HCMV-gB IgG response in humanized mice after: i.) sole iDCgB immunization (iDCgB, n = 4), ii.) iDCgB immunization followed by HCMV infection (iDCgB/HCMV n = 4, sorting data available for n = 3), or iii.) sole HCMV infection (HCMV n = 4), (**Fig 4A**, see gating strategy in **S5 Fig**).

Mice selected for these analyses contained elevated frequencies of IgG$^+$ B cells (**Fig 4B**, **full data S5 Fig**) and gB-reactive B cells (**Fig 4C**). Heavy chain Ig (IgH) variable regions from single B cells were amplified by PCR as previously described[22, 23]. All amplicons were sequenced, quality controlled, and annotated with IgBLAST [24] (**S6 Fig**). In total, 908 IgH productive sequences were analyzed to characterize the gB-specific antibody response (iDCgB, n = 399, iDCgB/HCMV n = 290 and HCMV n = 219) (**Fig 4D**). In each cohort, antibody characteristics were similar to the human IgG repertoire (**Fig 4D and 4H**). These include: i.) the distribution of the heavy chain V and J gene segments (**Fig 4H**), ii.) the lengths of the complementarity-determining regions of the heavy chains (CDRH3, median length of 16 amino acids) (**Fig 4E**) and iii.) the $V_H$ germline identity (median identity about 95%) (**Fig 4F**, Summary in **Table 1 and Table 2**).

IgG1 was the most abundant IgG subclass among all groups. We observed a tendency for HCMV-infected mice to have higher proportions of IgG$_3$ compared to mice immunized with iDCgB (**Fig 4G**). Interestingly, IgG$_1$ and IgG$_3$ have also been described to be the most critical anti-cytomegalovirus subclasses in humans [25]. IgH sequence analysis did not reveal an

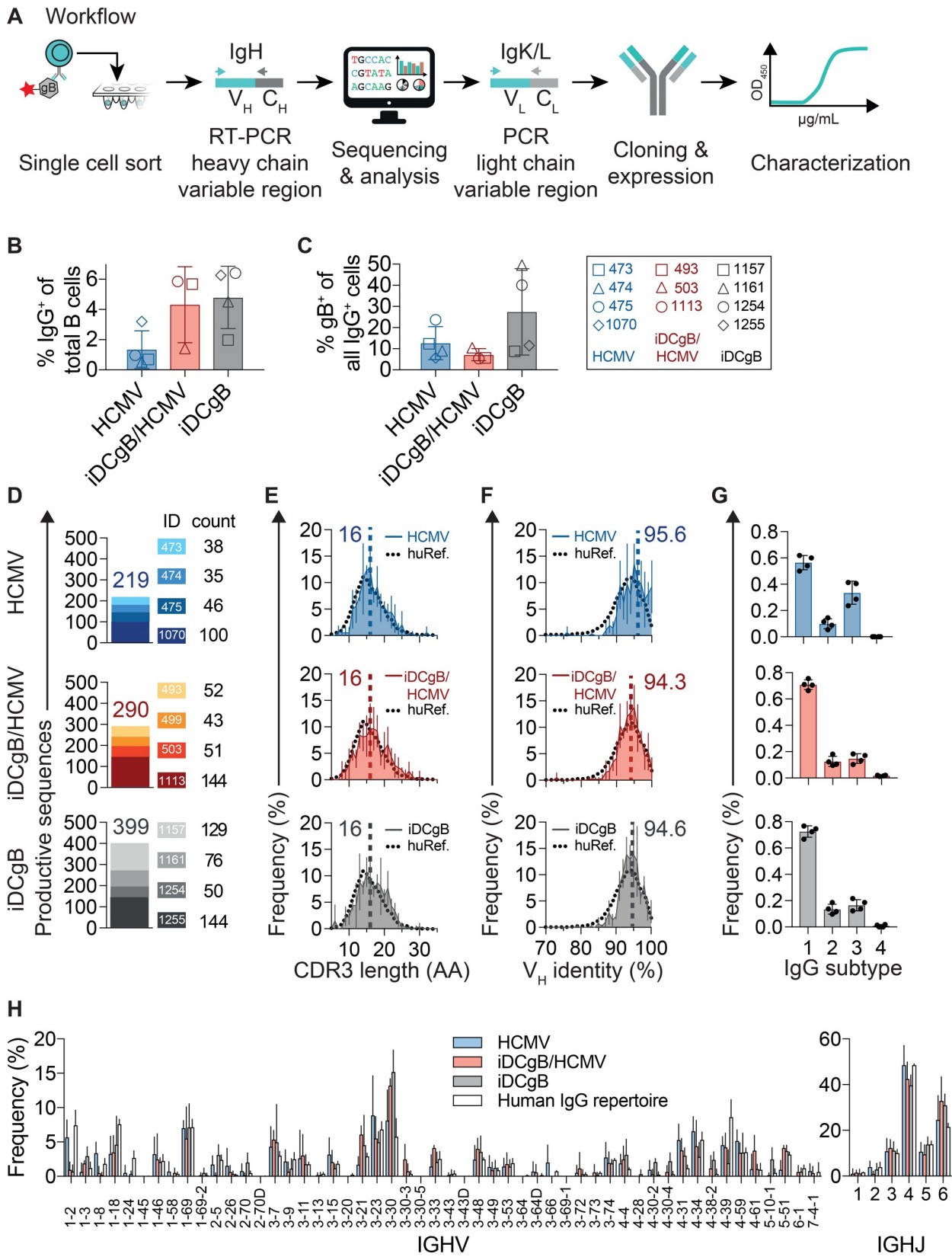

**Fig 4. Single B cell sorting followed by IgG sequencing revealed a broad B cell response. A)** Schematic representation of single cell sorting, PCR amplification of the IgG sequences, monoclonal antibody cloning and characterization workflow. Single gB-binding B cells were sorted into 96 well plates (gating strategy is shown in S3 Fig). Reverse transcription (RT) and heavy chain PCR amplification were performed at the single cell level. Products were sequenced by Sanger sequencing and annotated with IgBLAST for analysis. Selected light chains (Kappa or Lambda, i.e. IgK/L) were amplified by PCR, sequenced and annotated. Variable heavy ($V_H$) and variable light ($V_L$) chain pairs were sub-cloned into plasmids and recombinantly expressed in transfected HEK293-6E cells for functional characterization. Human reference data was taken from Erhardt et al. **B)** Frequency (% positive cells) of sorted $IgM^-/IgG^+$ B cells recovered for splenocytes obtained from HCMV (blue, n = 4), iDCgB/HCMV (red, n = 3) and iDCgB (grey, n = 4) cohorts. Error bars indicate standard deviation of the mean. **C)** Frequency (% positive cells) of gB-binding $IgM^-/IgG^+$ B cells is depicted (for each cohort as shown in b). Standard deviation of the mean is indicated. **D)** Numbers of quality-controlled and productive $V_H$ sequences amplified by nested-PCR. Each mouse identity (ID) is represented in a different color code and numbers of productive sequences obtained per mouse and per group are indicated. **E)** CDRH3 amino acid (AA) length distribution for all sequences per group. **F)** Mean $V_H$ germline identity (%) distribution for all sequences per group. Error bars in e) and f) depict standard deviations of the mean, dashed vertical lines and numbers indicate the median AA length and median $V_H$ germline identity, respectively. Human reference (huRef) distributions are indicated as black dotted lines. **G)** Distribution of IgG subtypes (1, 2, 3, 4) for all sequences per cohort. **H)** Mean IGHV and IGHJ gene segment usage for HCMV, iDCgB/HCMV, iDCgB cohorts and a human reference IgG repertoire. Error bars depict standard deviations of the mean. Human IgG repertoire reference (huRef) data in e), f), and h) was calculated using whole IgG NGS data from n = 4 healthy humans.

expansion of B cell clones. Taken together, iDCgB immunization alone or combined with HCMV infection induced a broad heterogenity of B cell sequences with characteristics similar to the human IgG repertoire.

## Functional analyses of recombinant mAbs generated from mice immunized with iDCgB

In order to evaluate antibody functionality, we generated nine mAbs from different mice using established methods [22, 23, 26] with minor modifications. Mice reconstituted with different $CD34^+$ donors were used (iDCgB: 4 mice generated from 2 donors; HCMV: 2 mice generated from one donor, iDCgB/HCMV: 3 mice generated from 2 donors, **Fig 4A**, **Table 1 and Table 2**). Nine recombinant antibodies containing unique sequences representing different mice and $CD34^+$ donors were readily generated (iDCgB: 4 mice generated from 3 donors; HCMV: 2 mice generated from one donor, iDCgB/HCMV: 3 mice generated from 2 donors, **Table 1 and Table 2**). The mAbs were tested for binding to recombinant soluble gB, binding to membrane-anchored gB present in 293T/gB cell lysates, and for their ability to inhibit TB40/GLuc infection of MRC-5 cells (**Fig 5A and 5E and 5I**).

As positive controls, human SM5-1 (a highly neutralizing mAb reactive against a discontinuous neutralization epitope on gB with a calculated half-maximal inhibitory concentration (IC50) against HCMV/TB40 of 2.2 μg/ml x $10^{-1}$) [27] and Kiovig (pooled human plasma) were used (**Fig 5B and 5F and 5J**). 3 out of 4 antibodies generated from iDCgB-immunized mice (DC06, DC14, DC16) showed binding to both soluble and membrane-anchored gB. 2 out of 4 antibodies (DC06 and DC16) neutralized HCMV infection (**Fig 5C and 5G and 5K**). 2 out of 3 antibodies selected from iDCgB-immunized/HCMV-protected mice bound to soluble gB (PR32 and PR17), or to membrane-anchored gB (PR32 and PR35) and 1 out of 3 neutralized HCMV infection (PR32). One antibody generated from HCMV-infected mice (CV4) showed binding to membrane-anchored gB, but neither bound to soluble gB nor neutralized HCMV (**Fig 5D and 5H and 5I**). mAbs generated from humanized mice immunized with iDCgB showed better performance than Kiovig but were out-performed by the high-affinity SM5-1($IC_{50}$ and $EC_{50}$ values and sequence annotations are in **Table 1**). The three neutralizing antibodies differed in their incorporated heavy and light chain V(D)J genes, their heavy chain CDR3 lengths (15–18 amino acids) and the number of somatic hypermutations (3–18 and 2–10 amino acid mutations for heavy and light chains, respectively). For the 10 monoclonal antibodies tested here (including the SM5-1 positive control) there was no evidence for a correlation between affinity/neutralization capacity and any sequence characteristic (see **Table 1**

**Table 1. Features of cloned and functionally tested monoclonal antibodies.** Monoclonal antibodies derived from iDCgB immunized mice (DC, n = 4), HCMV infected mice (CV, n = 2), and iDCgB immunized and protected against HCMV (n = 3). Control antibodies SM5-1 and KIOVIG are shown as references. CB unit = cord blood units used for humanization. EC50 binding capacity to recombinant gB protein (µg/ml) was calculated for clones DC06, DC16 and DC14 and DC16, PR28 and PR32. IC50 HCMV-neutralizing capacity (µg/ml) was calculated for clones DC06, DC16 and PR32. V, D and J genes depict top matches according to IgBLAST and IMGT nomenclature. Amino acid (AA) mutations include substitutions, insertions, and deletions. Characterictics of heavy chains are depicted.

| AB # | COHORT | SUBJECT | CB unit | EC50 gB binding (µg/ml) | IC50 NT (µg/ml) | VH_GENE | DH_GENE | JH_GENE | VH_IDENTITY | CDRH3_AA | CDRH3_AA_LENGTH | CH_ISOTYPE |
|---|---|---|---|---|---|---|---|---|---|---|---|---|
| CV03 | HCMV-Control | Mouse-473 | 346 | - | - | 3-30*18,3-30-5*01 | 3-10*02 | 6*03 | 98,3 | AKPVRGVYYSYYMDV | 15 | 3-1 |
| CV04 | HCMV-Control | Mouse-475 | 346 | - | - | 3-30*18,3-30-5*01 | 3-10*01,3-10*02,6-6*01 | 6*03 | 97,3 | AKPPRSGYYYYMDV | 15 | 3-1 |
| PR32 | iDCgB-HCMV | Mouse-1113 | 313 | 0.74 | 0,58 | 3-74*01 | 6-6*01 | 4*02 | 98,6 | QCRSIADRRSPEFDY | 15 | 2-1 |
| PR35 | iDCgB-HCMV | Mouse-499 | 346 | 0.05 | - | 3-48*02 | 5-18*01,5-5*01 | 6*03 | 90,5 | ARVTYSYGSPHQYYCMDV | 18 | 1-1 |
| PR28 | iDCgB-HCMV | Mouse-1113 | 313 | - | - | 4-4*02 | 3-10*01 | 5*02 | 94,9 | ARKPKVMVRGISDFFDP | 17 | 1-1 |
| DC06 | iDCgB | Mouse-1157 | 184 | 0.29 | 0,09 | 3-30*18,3-30-5*01 | 7-27*01 | 3*02 | 95,2 | AKPFASGWGWVNAFDI | 16 | 1-1 |
| DC14 | iDCgB | Mouse-1254 | 229 | 0.004 | - | 1-46*01 | 2-2*01,2-2*03 | 4*02 | 98,3 | ARGLRVYCSSSTCYATAGDY | 20 | 2-1 |
| DC16 | iDCgB | Mouse-1254 | 229 | 0.66 | 15,94 | 3-7*03 | 3-22*01 | 3*02 | 90,5 | VRDRNYHEGSTYYDVFDI | 18 | 1-1 |
| DC17 | iDCgB | Mouse-1255 | 229 | - | - | 1-46*01 | 4-17*01 | 4*02 | 91,6 | ARDHGDYHFFDS | 12 | 1-1 |
| SM5-1 | Reference | - | - | 0,09 | 0,014 | 1-2*02 | 2-8*01 | 3*01 | 91,9 | ARDGAKTVSNSGLSLLYYHNRLDA | 14 | - |
| Kiovig | Reference | - | - | 0,6 | 9,8 | - | - | - | - | - | - | - |

**Table 2. Characteristics of light chains.**

| AB # | COHORT | SUBJECT | CB unit | LIGHT_CHAIN_TYPE | VL_GENE | JL_GENE | VL_IDENTITY | CDRL3_AA | CDRL3_AA_LENGTH |
|------|--------|---------|---------|------------------|---------|---------|-------------|----------|-----------------|
| CV03 | HCMV-Control | Mouse-473 | 346 | Kappa | 3–15*01 | 4*01 | 98,9 | QQYNNWPS | 8 |
| CV04 | HCMV-Control | Mouse-475 | 346 | Kappa | 3–11*01 | 2*02 | 98,3 | QQRTNWPPECT | 11 |
| PR32 | iDCgB-HCMV | Mouse-1113 | 313 | Kappa | 3–20*01 | 2*01 | 98,6 | QQYGSIPYT | 9 |
| PR35 | iDCgB-HCMV | Mouse-499 | 346 | Kappa | 1–5*03 | 1*01 | 94,1 | QHRET | 5 |
| PR28 | iDCgB-HCMV | Mouse-1113 | 313 | Lambda | 1–40*01 | 3*02 | 93 | QSYDNSLSGSWV | 12 |
| DC06 | iDCgB | Mouse-1157 | 184 | Lambda | 3–21*02 | 3*02 | 95,8 | QVWDSSSDHVV | 11 |
| DC14 | iDCgB | Mouse-1254 | 229 | Kappa | 1–39*01,1–39*01 | 1*01 | 93 | QQSYSGPRT | 9 |
| DC16 | iDCgB | Mouse-1254 | 229 | Kappa | 1–6*01 | 1*01 | 94,8 | LQDFDYPRT | 9 |
| DC17 | iDCgB | Mouse-1255 | 229 | Lambda | 2–23*02 | 2*01,3*01 | 92,7 | CSYASMYTVV | 10 |
| SM5-1 | Reference | - | - | Lambda | 1–51*01, 1–51*02 | 2*01, 3*01 | 95,2 | GTPDRSLSV | 8 |
| Kiovig | Reference | - | - | - | - | - | - | - | - |

and **Table 2**). In summary, recombinant mAbs generated from mice immunized with iDCgB were able to bind to gB and neutralize HCMV *in vitro*.

## Passive immunization with mAbs

Ultimately, we evaluated in a proof-of-concept experiment if the best performing mAbs selected from the *in vitro* assays described above could provide *in vivo* protection against HCMV. In order to assess the effects of the mAbs on the HCMV infection, the antibodies DC06 and PR32 were administered i.v. (1 mg each) 16 weeks after HCT. For testing the effects against infection ("INFEC", the mAbs were administered a day prior to HCMV/GLuc challenge and then weekly for three consecutive weeks (**Fig 5M**; see core data and statistics in **S10 Table**). At this time-point, optical imaging analyses were performed comparing non-treated HCMV-infected controls (CTR/HCMV; n = 3) with mAb-treated mice (mAbs/HCMV n = 3). Compared with the controls, mice receiving passive immunization showed significantly lower intensities of the bioluminescent signal (P = 0.0294; 62.8% relative to median values obtained for controls) (**Fig 5N and 5O**; **S10A Table**). In a second step of the experiment, we evaluated if passive immunization could sustain an immune protection against HCMV reactivation when administered concurrently with G-CSF. At the endpoint of the experiment, two of the passively immunized mice sustained low levels of HCMV (average flux $1.9 \times 10^3$ p/s) whereas one mouse showed viral rebound (flux $3.5 \times 10^3$ p/s). Therefore, the passive immunization during HCMV reactivation yielded less pronounced and more variable results (P = 0.348; 65.9% relative to median values obtained for controls; **Fig 5N and 5O**; **S10B Table**). Although not statistically significant (P = 0.303) quantification of the viral copies in liver after reactivation showed that the passively immunized group retained only 14% of the PCR signal relative to the median values obtained for controls (**Fig 5P**; **S10C Table**). For analyses of BM, two out of three mice sustained low levels of viral copies but one mouse showed a HCMV rebound (P = 0.469; 32% reduction of median values obtained relative to controls; **Fig 5P**; **S10D Table**).

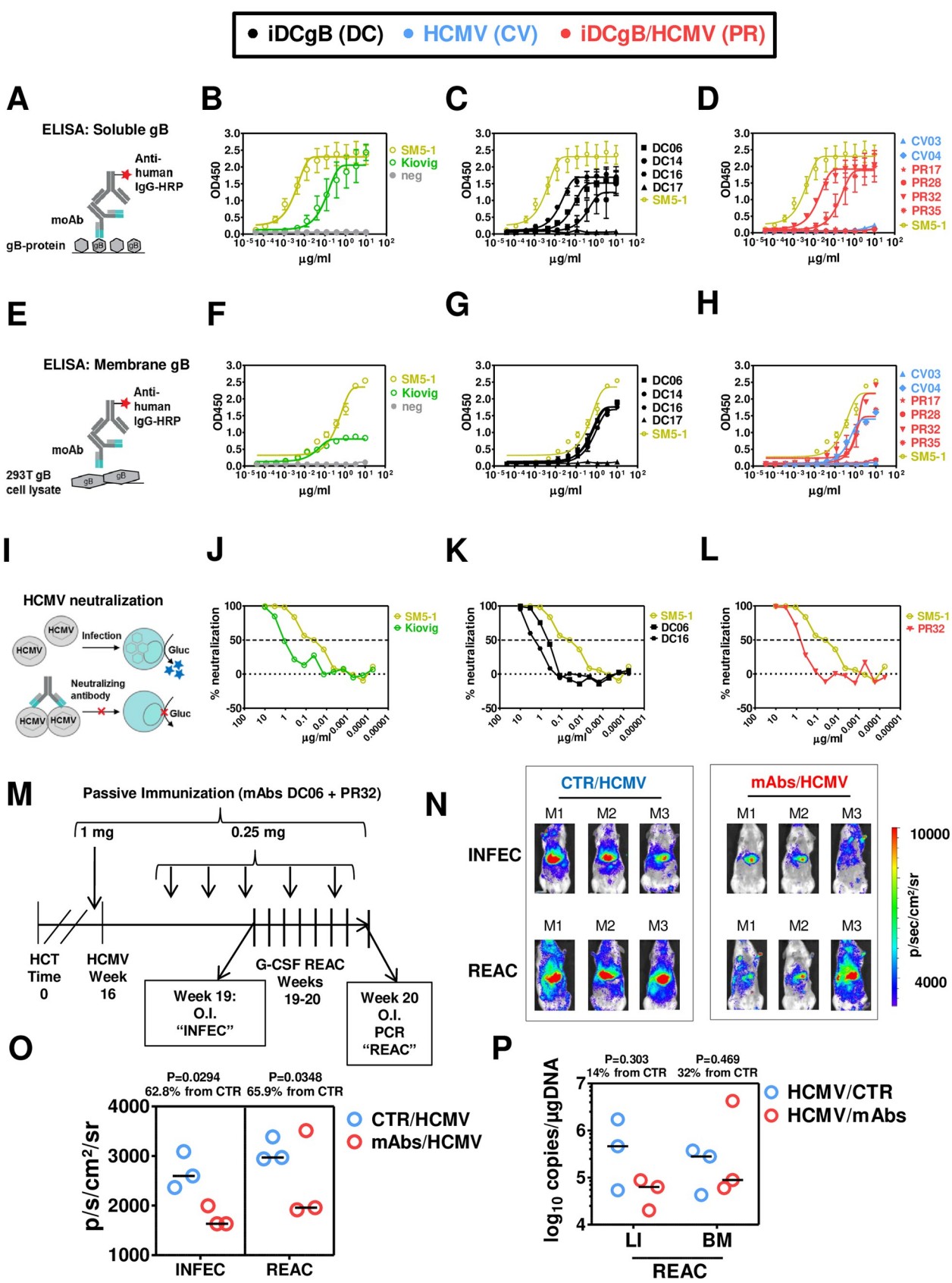

**Fig 5. Functional assessment of cloned monoclonal antibodies to bind to HCMV-gB and to neutralize HCMV *in vitro*. A)** Scheme of ELISA assay using wells coated with recombinant gB protein. Monoclonal IgGs were serially diluted, transferred to wells and immune detection was performed with an HRP-conjugated antibody against human IgG. Assays were performed in duplicate independent experiments and the results were merged for plotting the data. **B)** OD450 measurements for reference SM5-1 monoclonal IgG (orange), Kiovig pooled IgG (green) and negative control (grey). **C)** OD450 measurements for monoclonal IgGs derived from iDCgB immunized mice (black) are shown. SM5-1 was included as a reference (orange). **D)** OD450 measurements for monoclonal IgGs derived from HCMV (blue) and iDCgB/HCMV (red) cohorts are shown. SM5-1 was included as a reference (orange). **E)** Scheme of ELISA assay using wells coated with protein lysates from 293T/gB cells. Monoclonal IgGs were serially diluted, transferred to wells and immune detection was performed with an HRP-conjugated antibody against human IgG. Assays were performed in duplicate independent experiments and the results were merged for plotting the data. **F)** OD450 measurements for reference SM5-1 monoclonal IgG against HCMV gB (orange), Kiovig pooled IgG (green) and negative control (grey). **G)** OD450 depicted for monoclonal antibodies derived from iDCgB immunized (black) cohort. SM5-1 (orange) measurement was included as a reference. **H)** ELISA (OD450) measurements for monoclonal IgG derived from HCMV (blue) and iDCgB/HCMV (red) cohort. SM5-1 (orange) was included as reference. For **F) G) H)** Wells coated with protein lysates obtained from control 293T/w.t. cells were included in the ELISA assay as negative control. No cross-reactivity was detectable. **I)** Scheme of the *in vitro* neutralization assay. Antibodies were serially diluted and incubated with HCMV prior to infection. TB40-GLuc viruses were pre-incubated with the antibodies for 1 h and MRC-5 cells were infected with the virus-antibody mixture. Spinoculation was performed and 1h later medium was exchanged. The catalytic activity of the secreted GLuc signal was measured in the supernatant 24h later by luminometry as relative light units (RLU). The experiment was performed as independent duplicates and the results were merged. The % neutralization (y-axis) was plotted against the $\log_{10}$ µg/ml antibodies (x-axis), the neutralization curves were adjusted after non-linear regression analyses and the $IC_{50}$ values were calculated (**Table 1**). **J)** % neutralization of HCMV by the reference SM5-1 IgG (light green) and Kiovig polyclonal antibodies (dark green). Infected MRC-5 cells not exposed to the mAbs were included for RLU measurements and used as 100% infection reference (Mock, grey). **K)** % neutralization of HCMV by mAbs DC06 and DC16 derived from the iDCgB cohort (black). SM5-1 IgG (light green) is shown as reference. **L)** % neutralization of HCMV by the mAb PR32 derived from a HCMV-protected mouse of the iDCgB/HCMV cohort (red). SM5-1 IgG (light green) is shown as reference. The other six cloned antibodies (DC14, DC17, CV03, CV04, PR17, PR28 and PR35) did not neutralize HCMV infection. All assays were performed in duplicate independent experiments. For ELISA and neutralization assays, measurements were taken for two and four wells, respectively. For all graphs error bars indicate standard deviation of the mean and corresponding $EC_{50}$ and $IC_{50}$ values are shown in **Table 1**. **M)** Schematic representation of the experiment to test the effects of passive immunization by adoptive antibody transfer against HCMV. For the control group, mice were infected with HCMV 16 weeks after HCT and G-CSF treatment was performed between weeks 19 and 20 after HCT. For the passive immunization cohort, mice were injected i.v. with the monoclonal antibodies (DC06 and PR32, 0.5 mg each in total 1.0 mg) on the day prior to HCMV infection. After infection, the mice were injected i.v. with the monoclonal antibodies (DC06 and PR32, 0.125 mg each in total 0.25 mg) additional seven times. **N)** Optical imaging analysis performed for the control HCMV cohort (left) or the cohort treated with mAbs (D06 and PR32) to test the effects of passive immunization against HCMV infection (top, "INFEC") and after HCMV reactivation induced by G-CSF treatment (bottom, "REAC"). All mice were analyzed with the same settings; the range of the bio-luminescence signals is indicated by the colored bar on the right side (radiance: p/sec/cm$^2$/sr). **O)** Quantified total Flux (phonons/second, p/s) for the control (CTR/HCMV, blue) or mAb-treated cohort (mAbs/HCMV, red). ROI was quantified for the frontal torso and abdomen and kept constant for all mice. Data was obtained for the two time points to monitor INFEC (left) and REAC (right). **P)** Quantification of HCMV viral copies ($\log_{10}$ copies/ µg DNA) detected in liver (LI) and bone marrow (BM) comparing HCMV-infected mice (blue) and mice treated by with mAbs (red). The horizontal bars in black indicate the median values for each cohort and time of analyses. The P values were determined by t-Test. The % increase is relative to the CTR non-treated cohort (see **S10 Table**).

In conclusion, passive immunization with mAbs provided some level of protection against HCMV infection, but this protection was lessened upon HCMV reactivation.

## Discussion

Novel vaccines that can block or minimize HCMV infections and establishment of a latent res- ervoir providing a long-lasting immune control against HCMV are still sought as no vaccine candidate has been approved so far (reviewed by Reeves *et al.* [9] and Plotkin *et al.* [28]). Sev- eral vaccines candidates have explored HCMV-gB as viral antigen, such as a subunit vaccine (gB/MF59 [29, 30]) or vaccines based on the modified Vaccinia Ankara virus (MVA [31–36]). Results from pre-clinical testing in non-human animal models or from clinical trials showed induction of gB-specific and neutralizing humoral responses with improved clinical outcomes for vaccinated patients, but failed to fully protect against HCMV reactivation [9, 33, 34].

Efficacious immunization of immune compromised or immune deficient subjects is an unsolved medical riddle. In fact, the delayed B cell reconstitution observed in immune com- promised patients after HCT is a major limitation for vaccines seeking to promote humoral responses [37]. Here, we tested the protective effects of iDCgB in the HCT setting as an advanced therapeutic medicinal product (ATMP) as they consist on genetically engineered monocytes. In the European Union, testing of ATMPs require proof-of-concept in relevant *in*

*vivo* models and therefore we opted for a fully humanized system. Production of donor-derived iDCgB was straightforward for administration at early time-points after HCT into HIS mice. Vaccination significantly accelerated T and B reconstitution and expansion in different tissues, promoted gB-specific effector T cell and humoral responses.

The prophylactic iDCgB vaccination in the model resulted in a full protection against HCMV primary infection in the majority of the HIS mice. In the strict sense, the effects of iDCgB immunization to mitigate latency and/or to prevent HCMV reactivation remain to be further evaluated in longitudinal or cross-sectional experimental models. In the broad sense, iDCgB administered prior to a HCMV primary infection, showed marked effects in the development IgG and IgA class-switched B cells. Bioinformatics analyses of large datasets predicted IgG, IgA and CD4$^+$ T cell as key biomarkers of response accompanying iDCgB immunization. Interestingly, these markers have also been associated with protection against HCMV in humans in keeping the infection dormant [38, 39]. Further, B cell depletion studies designed to test the functional effects of B cells in the immune responses induced by iDCgB-immunization showed a modest viral rebound, indicating a possible relevance of the anti-gB humoral responses to maintain control against reactivation of latent HCMV reservoir. A major limitation of this pilot study using a clinical anti-CD20 mAb was that it was performed with small cohorts of mice. Nonetheless, it provided a useful experimental approach to be further developed to address the possible clinical effects of anti-CD20 therapies in jeopardizing the immune control of the silent HCMV latency, which could then progress towards viral reactivation.

Remarkably, iDCgB pre-immunized and protected HIS mice showed lower frequencies of PD-1$^+$ T cells. This implied that gB-specific responses spared T cells from HCMV chronicity-generated over-activation or exhaustion, previously observed in our HIS/HCMV model [16] and seen in patient-based studies [40]. Further, HCMV has been implicated with the onset and severity of autoimmune diseases, for instance rheumatoid arthritis (RA) or systemic lupus erythematosus (SLE) [41] and autoantibodies can play an essential role in disease progression [42, 43]. Using immune fluorescence analyses, we showed that the majority of the HIS mice protected against HCMV reactivation after iDCgB immunization rarely developed auto-reactivity, whereas non-protected mice showing HCMV reactivation showed frequently signs of humoral auto-immunity. Therefore, it is tempting to speculate that the iDCgB immunization could lower the onset of auto-reactive antibodies after transplantations.

The drawback of this *in vivo* HCMV model is the requirement of high levels of expertise and resources for generation of HIS mice and the fact that each experiment takes six months for completion. The conundrum is that HCMV infection is only consistently observed when applied after fifteen weeks after HCT, when human hematopoietic reconstitution in the mice is stable, robust and systemic. Therefore, more complex longitudinal or cross-sectional analysis of HIS mice are very demanding as several HCTs have to be established in parallel in order to allow statistical power of analyses.

For the current studies, we opted to evaluate the effects of prophylactic vaccination with iDCgB prior to a HCMV primary infection, because we were primarily interested in the early immune reconstitution and antiviral responses shortly after HCT. Further, a potent immunization against a primary infection could help to mitigate the establishment of a HCMV latent reservoir. However, from the clinical perspective, the highest risk group is when the HCT donor is HCMV-seronegative and the recipient is seropositive. In this case, the latent viral reservoir of the recipient is intrinsic. In this more complex scenario, the pre-existing latent viral reservoirs can disseminate during the early stages of immune reconstitution, when the recipient is immune compromised, causing serious HCMV disease. Therefore, there is an important need and a large interest to develop novel immunotherapeutic strategies to eradicate HCMV latent infections or to prevent the latent infection to reactivate (recently reviewed [44]). Since

during latency the viral gene expression is largely suppressed, indeed it would more challenging to expect immune responses against latency-associated genes [44]. Remarkably, cytotoxic CD4$^+$ T cells producing IL-10 and directed against the latency-associated proteins LUNA and UL138 have been detectable in healthy HCMV-positive donors [45]. However, these T cell responses directed to latency-associated antigens seem to be skewed to become immunosuppressive, and this is an immune escape mechanism that could potentially sustain viral latency.

The therapeutic effects of iDCgB Immunization after establishment of HCMV latency remains to be evaluated (therapeutic setting). To this end, mice would be infected 15 weeks after HCT and immunized with iDCgB at a later time-point. It would be interesting to evaluate if reactive T cells directed against lytic- or latency-associated antigens could be generated. In fact, using humanized mice infected with HCMV at 17–19 weeks after HCT, we recently showed therapeutic effects of adoptive T cells expressing an engineered chimeric antigen receptor targeting gb (gB-CAR-T) administered 8 weeks later [46].

From the translational perspective, the generation of lentivirus-induced iDCgB from a HCMV seronegative HCT donor under good manufacturing practices (GMP) for clinical use is straightforward. Upscaling and GMP-compliant standardized production of lentiviral vectors for generation of different modalities of iDCs has been achieved in our laboratory [47, 48]. A 48 h standard operating procedure (SOP) for cell manipulation has been established [48] and we are currently adapting this SOP towards a fully automated cell production system. The quality control analytical parameters for documentation of the ATMP cell product identity after thawing and characterization of the potency of IDC after culture were well defined and accepted by the European Medicines Agency (EMA). In the current study, we used integration-competent lentiviral vectors (ICLV) for iDCgB engineering, but for a clinical development we would prefer to use integration-defective (IDLV) as this vector system results into only residual genomic integration in transduced cells. Long-term studies (> 33 weeks) of iDCpp65 immunization in HIS mice showed no development of malignancies or exacerbation of acute graft-versus-host disease [20]. Thus, the promising prophylactic results obtained with the iDCgB vaccine in this HIS/ HCMV model could be translated clinically to the solid organ transplantation setting, when HCMV seronegative patients are on the waiting list and very likely will be transplanted in a near future with organs from seropositive donors. The data also warrant the testing iDCgB in the therapeutic setting, i.e. to predict if high-risk HCMV seropositive HCT patients transplanted with stem cells from seronegative donors (including cord blood donors) could be protected.

A remarkable corollary finding of our studies that opened space for development of another modality of immunization was the forthright discovery and production of gB-binding monoclonal antibodies directly from iDCgB immunized HIS mice. It is generally assumed that HIS mice have an impaired human B cell development and class-switch and fail to producing human-like humoral responses [13, 49]. But this notion is starting to vanish in the community of scientists improving humanized mouse models. DiSanto and collaborators generated a Balb/c $Rag2^{-/-}Il2rg^{-/-}SirpaNOD$ (BRGS) transgenic strain co-expressing the thymic-stromal-cell derived lymphopoietin (TSLP) [50]. The resulting BRGST-HIS mice generated IgG$^+$ mature B cells and enhanced humoral responses against Keyhole limpet hemocyanin (KLH). Danner et al showed that NRG mice expressing HLA class II (HLA-DR4, DRAG) transplanted with DR-matched HSCs developed functional B cells and IgG responses upon tetanus toxoid vaccination [51, 52]. Cavaccini and collaborators showed that HIS mice infected with human immune deficiency virus-1 (HIV-1) generated a broad variety of Igs. Using splenocytes, hybridomas were cloned and mAbs displayed anti-HIV binding and neutralization capacity, providing a library of isotypes and gene family usage [53]. Flavell and collaborators showed that human IL-6 knock-in in HIS mice transplanted with human fetal tissues produced OVA-

specific B cells. Single CD10⁻CD27⁺IgG⁺ B cells from spleens of immunized mice were OVA-specific [54].

To our knowledge, our current work is the first demonstration showing that human mAbs with potent neutralizing activity could be directly and efficiently cloned from human B cells obtained from HIS mice, bypassing the need of B cell immortalization or hybridoma generation. We demonstrated that iDCgB immunization enabled development of high frequencies of IgG⁺ gB-binding B cells. These B cells were efficiently sorted at the single cell level, with subsequent IgG sequence amplification, molecular characterization and production of recombinant monoclonal antibodies. The discovered isolated antibody sequences resembled the human IgG repertoire and were mainly members of the $IgG_1$ class [55]. Three of the four cloned and tested mAbs generated from iDCgB mice were gB-specific and three were able to neutralize HCMV *in vitro*. Although their neutralizing activities were lower than the activity of the remarkable, highly neutralizing human mAb SM5-1 [21], they showed higher activity than Kiovig, a clinically used pooled polyclonal IgG product obtained from human donors [56]. Neutralization capacity of monoclonal antibodies targeting viral infections is often thought to be the primary *in vivo* function. However, effector functions of antibodies associated with protection *in vivo* are likely to be complex and it is still controversially discussed if neutralization titers strictly correlate with protection against HCMV [57, 58]. Nelson *et al.* [29] reported for instance that a subunit vaccine (gB/MF59) was able to induce antibody-dependent cellular phagocytosis (ADCP) responses by inducing an $IgG_3$ response towards the membrane-bound antigenic domain 3 (AD-3) of gB. In addition, due to their strong Fc receptor binding, these antibodies also induced antibody-dependent cellular cytotoxicity (ADCC).

Most likely, combinations of different mAbs with different specificities and immunologic properties would be the best clinical strategy to bypass HCMV immune evasion or mutational escape. This is highlighted by the evidence presented herein that two of the mAbs selected on the basis of high gB-binding capacity, when used for passive immunization of HIS/HCMV mice, showed potency against HCMV infection and were even able to endure viral reactivation. Those are promising results from our prototypic proof-of-concept experiments. A larger high-throughput screening of several mAbs will be needed to select other highly diverse anti-gB IgGs and also IgAs for future more in depth *in vivo* potency analyses in HIS/HCMV mice in order to select clinically relevant lead candidates or combinations of them.

Further, the iDC platform can evolve to combine the expression of additional HCMV antigens, such as the trimeric (gH, gL, gO) or pentameric (gH, gL, UL128-131A) glycoprotein complex to induce antibodies with broad and potent neutralization capacity [10, 59–61]. HCMV cell-to-cell transmission is an important viral spread mechanism considered to play an important role in wild-type HCMV strains, which could be potentially troublesome to counteract with antibodies targeted against gB. Whether mAbs against gB developed with our technology will have an effect on the cell-to-cell spread *in vivo* remains to be shown. Cell-to-cell spread may be better prevented with antibodies targeting the pentameric complex [60]. One could overcome such limitations once these antigenic complexes are expressed altogether and assembled on the surface of the iDC to promote antibodies that inhibit the HCMV cell-to-cell spread. This would then enable the cloning of mAbs targeted to non-contiguous cis-epitopes or epitopes, and possibly even of mAbs targeted against epitopes in different proteins and presented *in trans* to the BCR.

In conclusion, the current work underscored that: (i) HIS/HCMV models can be used for preclinical testing of active and passive immunization approaches against HCMV; (ii) exploring these models, we demonstrated preclinical potency of iDCgB as a vaccine modality aimed at concurrently promoting human immune reconstitution and immunizing against gB for stimulation of gB-specific T and B cell responses; (iii) mAbs against HCMV-gB could readily

be generated using the HIS/iDCgB platform and this strategy can further evolve to generate a broad range of novel fully human antiviral neutralizing mAbs for the biotechnology sector and for clinical translation.

## Material and methods

### Ethics statement

Umbilical CB units were obtained of the mothers in accordance to study protocols approved by Hannover Medical School Ethics Review Board under written informed consent (approval Nr. 4837). Cord blood was obtained from mothers without complications such as acute HCMV infection or reactivation during pregnancy. Neonates were born at term and did not present clinical signs of fetal or congenital HCMV infection. Mice experiments were performed in accordance with the regulations and guidelines of the animal welfare of the State of Lower Saxony ("Niedersächsisches Landesamt für Verbraucherschutz und Lebensmittelsicherheit, Dezernat 33/Tierschutz," approval Nr. 33.12-42502-04-1). Mice euthanasia was performed by cervical dislocation after $CO_2$ narcosis.

### Cell culture and primary cells

MRC-5 (human fetal lung fibroblasts, ATCC, Manassas, VA, USA) and HEK-293T cells (human embryonic kidney cells, ATCC) were expanded and cultured in D10 medium (containing DMEM (ThermoFisher, Waltham, MA) containing 10% fetal bovine serum (FBS, HyClone, Logan, UT), 1% non-essential amino acids (NEAA, for MRC-5 cells) (Thermo-Fisher) and 1% penicillin/streptomycin (Merck Millipore, Billerica, MA) at 37˚C with 5% $CO_2$. Peripheral blood mononuclear cells (PBMNC) were isolated by Ficoll gradient centrifugation as described before (19). Immune magnetic separation was used to isolate CD14+ and CD34+ cells (Miltenyi Biotech, Bergisch-Gladbach, Germany) as described before [19].

### Production of lentiviral vectors

The HCMV-gB open-reading frame was inserted into the RRL self-inactivating lentiviral transfer vector using routine PCR and sub-cloning techniques [62]. Third-generation integrase-competent lentiviral vectors (ICLV) co-expressing GM-CSF and IFN-α and expressing HCMV-gB were produced and titrated by quantifying the p24 HIV-I core protein by ELISA, as previously described [18]. Additionally, a tricistronic vector was generated and used co-expressing GM-CSF, IFN-α and HCMV-gB.

### HCMV propagation, titration and generation of MRC-5 infected cells

HCMV-GLuc seeding stocks were generated using a TB40-BAC4-cloned HCMV variant and propagated on MRC-5 cells as described [63]. Briefly, cell-free supernatants were concentrated by centrifugation at 19,000 rpm for 2 h 20 min and stored at −80˚C. Viral batches were titered as described [16]. For the generation of batches of HCMV-infected carrier cells, $1.0 \times 10^7$ MRC-5 cells were infected at a multiplicity of infection (MOI) of 1 overnight at 37˚C, the medium was replenished and then the cells were cultivated for additional 2 days to produce infected cells 3dpi. Cells were detached and cryopreserved in freezing medium containing 15.5% human albumin, 10% DMSO (Braun, Melsungen, Germany), and 5% Glucose (Sigma-Aldrich, St.Louis, MO). Quality control was performed with mouse anti-human HCMV-gB p27-287 antibody and a secondary anti-mouse-IgG Alexa647 antibody (Biolegend, San Diego, CA). The frequency of infected (gB positive) cells was determined by flow cytometry using an LSR II cytometer (BD Bioscience, Becton Dickinson GmbH, Heidelberg, Germany).

## Generation of iDCgB and quality control

The same CB unit used as a source of CD34[+] cells for mouse HCT was used for iDCgB production. For the generation of iDCgB, CD14[+] monocytes were isolated from CB using immune magnetic beads (Miltenyi Biotec). After isolation, monocytes were pre-conditioned with recombinant human GM-CSF and IL-4 (both 50 ng/ml; Cellgenix, Freiburg, Germany) for 8 h at 37˚C in X-Vivo media (Lonza, Basel, Switzerland). Afterwards, media was removed and LVs were added plus 5 µg/ml protamine sulfate (Valeant, Duesseldorf, Germany). Co-transduction was performed with LV-GM-CSF/ IFN-α (2.5 mg/ml p24 equivalent) plus LV-HCMVgB (3.0 mg/ml p24 equivalent) for 16h. Afterwards, cells were washed extensively with 1xPBS and cryopreserved. For quality control, an aliquot of the batch was thawed and cultured for 7 days in X-Vivo media at 37˚C. Cells were analyzed by flow cytometry for the DC differentiation markers (CD80, CD86 and HLA-DR) and for the surface expression of HCMV gB (for antibodies see S7 Table). For immunizations, cells were thawed, washed, re-suspended in PBS, recounted and administered.

## HCT, iDCgB immunization, HCMV infection and reactivation

NRG mice were obtained from The Jackson Laboratory (JAX, Bar Harbor, ME) and bred in-house under pathogen-free conditions. Female mice were used for all experiments (19). Briefly 5 weeks-old mice were sub-lethally irradiated (450 cGy) using a [$^{137}$Cs] column irradiator (Gammacell 3000 Elan; Best Theratronics, Ottawa, Canada). Four hours after irradiation, $2.0 \times 10^5$ human CB-CD34[+] cells were injected as described before for HSCT [19]. CB units were tested prior to experiments for humanization potential. Only CB units which resulted into >20% hCD45 (BL) at week 10 post-HCT were used in studies. Cryopreserved and quality-controlled iDCgB were thawed and injected into NRG at weeks 6, 7, 10 and 11 after HCT. Mice were immunized with a total dose of $5.0 \times 10^5$ iDCgB cells injected s.c., near the anatomical regions of the inguinal and axillary lymph nodes. Humanized NRG mice were infected with HCMV as described before (16). Briefly, at week 17 post HCT, mice were treated s.c. with 150 ng hG-CSF/mouse/day (Granocyte, Kohlpharma GmbH, Merzig, Germany) for 3 days, injected i.p. with $1.0 \times 10^6$ MRC-5 cells previously infected with TB40-GLuc (MOI 1) and treated for G-CSF for additional 2 days. For HCMV reactivation, mice were treated s.c. with 2,5 µg hG-CSF/mouse/day for 7 days from weeks 24 and 25 post-HCT.

## Bioluminescence *in vivo* optical imaging

Mice were anesthetized with Ketamine/Xylazine (10 and 5 mg/kg, respectively, Sigma-Aldrich). Coelenterazine (Promega, Madison, WI) was solubilized in ethanol (5 mg/ml) and diluted shortly prior to administration in PBS (final dose 50 µg/mouse). Mice were injected i.v. with 100 µl of coelenterazine solution and imaged immediately after administration. *In vivo* optical imaging analyses were performed with an IVIS SpectrumCT (PerkinElmer, Waltham, MA) and all pictures were taken with a field of view C, f stop 1 and medium binning for each mouse. Exposure time was set to 300 seconds per mouse. The anatomical region of interest (ROI) was kept constant for quantified analyses of all mice and p/s were calculated. Data analysis was performed with Living Image Software (Perkin Elmer).

## RT-q-PCR for detection of HCMV viral copy number

DNA was isolated from tissues with the Blood and Tissue isolation kit (Qiagen, Hilden, Germany). The corresponding DNA concentration was determined by spectrophotometry and stored in aliquots at −20˚C. Artus CMV TM PCR kit (Qiagen) was used to determine HCMV

viral copies according to the manufacturer's instructions. The standard curve, negative and internal controls were provided by the kit. PCR was performed and analyzed with StepOne-Plus-PCR cycler (ThermoFisher). The Ct values were used to calculate the copies of HCMV genomes per sample adjusted with the respective DNA concentrations.

## Flow cytometry

Cells recovered from spleen and blood were incubated with a hypotonic solution (0.83% ammonium chloride/20 mM HEPES, pH 7.2, for 5 min at RT) to lyse the erythrocytes. Blood and tissues were further processed as described before for immune staining and flow cytometry analyses [18]. Cells were incubated with prior titrated antibody concentrations. Cells were washed to remove unspecific antibodies. Data was obtained with a LSR II cytometer (BD Biosciences). Antibodies against all markers are provided in **S7 Table**.

## Intracellular cytokine staining

Cryopreserved cell material obtained from mLN were thawed and washed once to remove remaining DMSO. mLN from each experimental group from each individual reconstitution were pooled together for analysis. Subsequently, cells were resuspended in TexMACS media (Miltenyi Biotec) supplemented with 3% heat-inactivated human serum (c.c.pro, Oberdorla, Germany) and 5 ng/ml IL-7 and IL-15 (Miltenyi Biotec). $5x10^5$-$1x10^6$ cells were seeded into each well of a 96 well U-bottom plate (Sarstedt). As control, PBMCs from anonymous healthy CMV-seropositive donors were freshly isolated from disposable platelet (PLT) apheresis routinely collected at the Institute of Transfusion Medicine and Transplant Engineering from the Hannover Medical School (MHH). Informed consent was obtained from all donors as approved by the Ethics Committee of Hannover Medical School. Cells were rested at 37˚C and 5% $CO_2$ for 4 h, followed by stimulation with 250 μg/ml of gB protein (provided by Marija Backovic), 1:25 dilution of pp65 protein (Lophius, according to kit manual in media), 1:25 dilution of IE1 protein (Lophius, according to kit manual in media), respectively. As positive staining controls, cells were stimulated with 10 ng/ml of phorbol 12-myristate 13-acetate (PMA; Sigma Aldrich) and 500 ng/ml of ionomycin (Sigma Aldrich) and unstimulated cells were used as negative controls. After 1 h of stimulation, 5 μg/ml of Brefeldin A (Biolegend) was added to each well and cells were carefully mixed by pipetting. Cells were incubated at 37˚C and 5% $CO_2$ for further 15 h. For staining, cells were harvested into 5 ml polystyrene tubes (Sarstedt) and washed once with PBS (Lonza) prior to surface staining. Afterwards, cells were fixed, permeabilized, and stained intracellularly using IntraPrep Permeabilization Reagent (Beckman Coulter, Brea, USA) according to the manufacturer's instructions. Antibodies used are listed in **S7**. Samples were acquired at FACSCanto with 10-color configuration (BD Biosciences) using FACSDiva 8.0.1 (BD Biosciences) and analyzed with FlowJo (Treestar Inc.).

## Detection of autoantibodies in serum of humanized mice by ANA immune fluorescence staining

Antinuclear antibodies (ANA) were measured using a commercially available assay (AESKUS-LIDES ANA-HEp-2; Aesku.Diagnostics, Wendelsheim, Germany). The sera obtained from the humanized mice were diluted 1:10 in sample buffer and a FITC-labeled goat-anti-human IgG (H+L chains) was used as a secondary antibody. In addition, anti-mitochondrial and anti-parietal cell antibodies were measured at a serum dilution of 1:10 using the kits EUROPLUS kidney (rat) and Stomach (monkey) with urea pretreatment (Euroimmun, Lübeck, Germany) and the FITC-labeled conjugate anti-human IgG which was provided in the kits. All

immunofluorescence tests were evaluated and photographed by a highly experienced technician using the fluorescence microscope Axiostar plus (Carl Zeiss, Göttingen, Germany).

## gB protein production

The synthetic gene encoding HCMV gB (GenBank: AGL96655.1) from the HCMV-TR strain [64] was codon-optimized for expression in *Drosophila melanogaster* Schneider cells (S2). The segment encoding residues 90 to 704 of the gB ectodomain was cloned into the pTß350 expression vector containing a Bip signal peptide at the N-terminus and a double strep-tag at the C-terminus for expression in S2 cells. Mutagenesis was performed to change the fusion loop residues and avoid protein aggregation by modifying the YAYIH[154-158] and GSTWLY[238-243] residues into EEEEE and EEEEEE, respectively. Generation of the S2 cell lines secreting the gB ectodomains and the protein production were performed as described[46]. The recombinant gB ectodomains were purified from the supernatant by affinity chromatography using StrepTactin resin (IBA Biosciences, Göttingen, Germany), dialyzed into PBS and used as such for the experiments. This protein production technique results in the assembly of the trimeric gB observed by size exclusion chromatography.

## Quantification of gB-binding by ELISA and HCMV *in vitro* neutralization assays

Antibody binding was assessed by two different ELISA assays using different antigenic targets: 1) Recombinant purified gB protein; 2) Cell lysates derived from 293T-wt and 293T-gB cells. Antigen coating: 96 well plates (Coring, Corning, NY) were coated with 2 μg protein or lysate diluted in 50 μl and kept overnight at 4˚C. Plates were washed 5x with PBS/0,1% Tween, blocked with 100 μl PBS containing 5% FBS for 2 h at 37˚C, washed 3x with PBS/0,1% Tween. Antibody binding: Antibodies were diluted in PBS / 2% FBS and each antibody dilution was assayed in duplicates. Plates were incubated for 2h at 37˚C. Afterwards plates were washed with 5x with PBS/0.1% Tween. Detection: 100 μl anti-human IgG conjugated with horseradish peroxidase (HRP) (1:1000 in PBS/2% FBS) were added per well, the plates were incubated for 45 min at 37˚C, washed 3x with PBS/0.1% Tween 100 μl of 3,3′,5,5′-Tetramethylbenzidin (TMB) (ThermoFisher) solution was added and reaction was stopped with 100 μl $H_2SO_4$ (Roth). OD450 was obtained with an ELISA plate reader (Molecular Devices). For the neutralization assays, $1.0x10^5$ MRC-5 cells were seeded in 96-well cell culture plates and cultured overnight 37˚C at 5% $CO_2$. Next, antibodies diluted in D10+ NEAA were added to TB40-GLuc (MOI 0.1) was added to pre-diluted antibodies (100 μl total) and incubated for 1h at 37˚C and 5% $CO_2$. Antibody/virus mix was added to MRC-5 cells, the plates were spinoculated for 30min at 1000xg at 32˚C and then incubated for 1h at 37˚C at 5% $CO_2$. The supernatant was aspirated and cells were washed once with D10+NEAA. Finally, 100 μl of fresh media was added to cells and incubated for 24 h at 37˚C at 5% $CO_2$. Next, 20 μl of cell supernatants containing secreted GLuc were transferred to wells of a white 96-well plate and bioluminescence was measured 3 s after automatic injection of 50 μl PBS containing 0.2 μg/ml coelenterazine (Promega, Madison, WI). Bioluminescence was measured in a plate reader (Tristar[2]; Berthold Technologies, Bad Wildbach, Germany).

## Luminex bead assay for detection of human cytokines and immunoglobulins

Human Th1/Th2 cytokines and IgG subtype analysis in plasma were quantified by fluorescent bead based 14-plex Luminex assay and concentration was determined (Darmstadt, Germany)

according to the manufacturer's instructions (Merck Millipore, Darmstadt, Germany). Human IgG and IgM ELISA kit (eBioscience, San Diego, CA) was used according to instructions written in the manual. Plasma samples were thawed on ice, diluted, and samples were measured in triplicates. ELISA plates were analyzed with Spectramax 340PC384 plate reader (Molecular Devices, Sunnyvale, CA) and optical density (OD) was calculated.

## Detection of HCMV specific IgG in plasma of humanized mice

For detection of HCMV specific IgG against several HCMV antigens (IE1, CM2, p150, p65, gB 1 and gB 2) recomLine CMV IgG kit was used according to according to the manufacturer's instructions (Mikrogen GmbH, Neuried, Germany). Photos were taken with a Chemidoc station (Bio-Rad, Hercules, CA).

## Labeling of gB protein with Alexa-647 fluorophore

Dialyzed gB protein (1 mg/ml) was labeled with Alexa Fluor 647 Protein Labeling Kit (Thermo Fisher), aliquoted and stored at -20˚C. Labeling and quality control analyses were performed according to the protocol provided in the kit.

## Flow cytometry cell sorting

For sorting of gB-specific B cells, cryopreserved viable splenocytes were thawed in pre-warmed PBS + 1% FBS. After centrifugation (300xg, 7 min) cells were blocked in 100 µl PBS + 10% FBS for 30min on ice and then washed once with PBS + 1% FBS. Next, cells were stained with fluorochrome-conjugated antibodies against human CD45, CD19, IgM and IgG (S7 Table) and with 0,5 µg/ml of Alexa-647 labeled HCMV-gB protein (described above) in 100 µl total volume. Cells were incubated for 30 min at 4˚C. 10 min before end of incubation time 7AAD was added to cells. Cells were washed once with PBS + 1% FBS and re-suspended in appropriate volume for sorting. Single $7AAD^-/CD45^+/CD19^+/IgM^-/IgG^+/HCMV\ gB^+$ cells were sorted with a FACSAria Fusion cell sorter (BD bioscience) into 96-well PCR plates (PEQlab/VWR, Darmstadt, Germany) which were filled with 4 µl sorting buffer consisting of 2 U/µl RNAsin (Promega), 1 U/µl RNAse OUT (Thermo Fisher), 10 mM DTT (Promega), 5% 10x PBS (Thermo Fisher) in Nuclease-free water, not DEPC treated (Thermo Fisher). Sorting was performed at low sample pressure differential using a single cell precision matrix. 96-well plates were covered with a cover foil (PEQlab/VWR) and stored at -80˚C.

## Antibody sequencing, analysis and production

Antibody repertoire analysis and cloning were performed as previously described [22, 26]. In brief, cDNA was generated from single B cells with Superscript IV (Invitrogen) and random hexamer primers (Thermo Fisher). Targeted amplification of IgG heavy and light chains was performed in a semi-nested PCR with IGHV-, IGKV- and IGLV-specific forward primer mixes [23], constant region-specific reverse primers [22,65] and Hotstart Taq polymerase (Invitrogen). PCR products were analyzed by Sanger sequencing with a constant region-specific reverse primer ('GTTCGGGGAAGTAGTCCTTGAC') [66]. Sequences were filtered for a mean Phred score of 28 and a length of at least 240 nucleotides. V(D)J annotation was performed with IgBLAST (24) and sequences were trimmed to the variable region from framework region 1 (FWR1) to the end of the J gene. Low quality sequences with more than 15 nucleotides with a Phred score below 16, stop codons, or frameshifts were excluded from further analyses. B cell receptors (BCR) with identical $V_H$ genes and a similarity of at least 75% in the CDRH3s (determined by the pairwise Levenshtein distance with respect to the shortest

CDRH3) were denoted as clonally related if they were derived from a single mouse or "shared" if they were derived from different mice. Selected antibody heavy and light chains were cloned into expression vectors by sequence- and ligation-independent cloning (SLIC) [26] and purified from supernatants of polyethylenimine (PEI)-transfected HEK293-6E cells with Protein G-Sepharose (Merck) as previously described [67]. Ehrhardt et al. was furthermore used to obtain human reference cohort for comparisions.

### *In vivo* B cell deletion with anti-CD20 antibody

NRG mice were humanized, immunized with iDCgB, infected with HCMV and treated with G-CSF as described before. For one experiment, iDCgB was generated after transduction with a tricistronic vector encoding for GM-CSF/INF-α and gB, which resulted into the same typical co-expression of DC markers and gB (**Fig 1A**). B cell immune-depletion was performed after G-CSF treatment for HCMV reactivation. All mice were treated i.v. with 10 mg/kg anti-CD20 monoclonal antibody (Rixathon, Holzkirchen, Germany). Optical imaging analyses were performed before depletion and 5 days after anti-CD20 treatment.

### Passive immunization of humanized NRG mice with recombinant antibodies

NRG mice were humanized as described before and challenged with HCMV at week 16 after HCT. For passive immunization, 1 day prior to HCMV challenge, mice were injected i.v. with 1 mg of DC06 and PR32 (0.5 mg of each antibody). After the challenge, mice received additional i.v. injections of 0.25 mg DC06 and PR32 (0.125 mg of each antibody) every 7 days for 3 weeks. From week 19 to 20 after HCT, 2.5 μg G-CSF were administered daily s.c. for 7 days. Passive immunization with 0.125 mg of each antibody was further administered on days 2 and 5 during G-CSF treatment. The experiment was terminated and mice were analyzed at week 20 post HCT.

### Statistical analysis

Data consisting of the percentage an absolute cell counts of human lymphocyte subsets in blood and tissues (# positive cells and % positive cells) were organized in a PivotTable using Excel software 2010 (Microsoft, Redmond, WA). The comparison of the two groups (HCMV vs. iDCgB/HCMV) for total cell counts was carried out by negative binomial regression using rate ratio, whereas beta regression using odds ratios was applied for the comparison with respect to the percentages. Negative binomial and beta regression were carried out with the open-source statistics software R (R Core Team 2018). P-values for all other comparisons were calculated using Welch's t-Test and the Wilcoxon-Mann-Whitney test. Significances are indicated and described in the figure legend. For calculation of differences between control mice and mice treated with mAbs (anti-CD20 or passive immunization), i.e. signal relative to baseline; we used the quotient of the medians.

### Bioinformatic analysis

Data configuration: All measurements for cell phenot2ypes were determined as relative frequency (%) and counts (#) in different organs (BM: bone marrow, LI: liver, LN: lymph nodes, mLN: mesenteric lymph nodes, SPL: spleen, SG: salivary glands, Thy: thymus). The dataset contained 29 samples (12 and 17 samples for reactivated and vaccinated groups, respectively). All measurements without missing values (185 biomarkers) were considered as input variables for the classification problem. Features selection: The two-sample Kolmogorov-Smirnov (KS)

test, a nonparametric test of the equality of one-dimensional probability distributions, was used to evaluate the difference between underlying probability distributions in reactivated versus vaccinated groups. The KS statistic (D), defined as the maximal vertical distance between the empirical cumulative distributions of reactivated and vaccinated groups, and significance level (p-value) were the two criteria considered for reducing the number of biomarkers for the classification problem. Classifier: Fuzzy rule-based systems (FRBS) are universal approximators of continuous functions based on linguistic variables and rules, which were used herein as classifier systems for predicting sample groups. FRBS are interpretable systems for data analysis and modeling tasks that account for uncertainty, imprecision, and non-linearity in complex systems analysis. Ishibuchi's method was used for creating FRBS from data [68], and implemented using the frbs package [69] for the statistical computing software R (version 3.4.3) [70]. Classification accuracy: Due to low sample size, a nested cross-validation (CV) pipeline was used to obtain unbiased classification accuracies (**Fig 3C**). In a main 10-fold CV loop, the KS test was performed and a subset of biomarkers was selected based on high KS statistic (D>0.3) and low p-value (p<0.1). Thereafter, all possible combinations of 2, 3 and 4 biomarkers from the selected subset were tested for classification in an inner CV loop. The biomarkers combinations with high average training ($A_{Train}$>85%) and inner CV classification accuracies ($A_{Inner}$>65%) were selected according to the number of occurrences in the main CV loop.

## Supporting information

**S1 Fig. Gating strategy for detection and quantification of T and B cells.** Post-transplant immunization of humanized NRG mice with iDCgB improved the maturation of T cells.
(TIF)

**S2 Fig. *In vitro* re-stimulation and gating strategy for detection of antigen-HCMV specific T cell responses and gating strategy for detection of follicular T helper cells.**
(TIF)

**S3 Fig. Immunization of huNRG mice with iDCgB prior to HCMV infections/reactivations promoted higher numbers and maturation of human T and B cells.**
(TIF)

**S4 Fig. Strategy for data classification between HCMV and iDCgB+HCMV and demonstration of B cell depletion.**
(TIF)

**S5 Fig. Gating strategy for sorting single IgM⁻/IgG⁺ gB-binding B cells.**
(TIF)

**S6 Fig. Single cell quality control and clonality analysis.**
(TIF)

**S1 Table. Descriptive statistics regarding immune phenotype of cells analyzed from SPL and mLN in total numbers and SPL in percentage.**
(DOCX)

**S2 Table. Descriptive representation of data shown in Fig 2 regarding the HCMV PCR analyses.**
(DOCX)

**S3 Table. Qualitative analyses for detection of human IgG-reactivity in plasma of mice by strips immunoassays.**
(DOCX)

**S4 Table. Descriptive statistics regarding immune phenotype of cells from BM in total numbers.**
(DOCX)

**S5 Table. Descriptive statistics regarding immune phenotype of cells from SPL in total numbers.**
(DOCX)

**S6 Table. Descriptive statistics regarding immune phenotype of cells from SPL in percentage.**
(DOCX)

**S7 Table. Monoclonal antibodies used in the studies.**
(DOCX)

**S8 Table. Descriptive classification data.**
(DOCX)

**S9 Table. Descriptive statistics regarding the *in vivo* B cell depletion experiments.**
(DOCX)

**S10 Table. Descriptive statistics regarding the passive immunization experiments.**
(DOCX)

## Acknowledgments

The authors are grateful for the excellent technical advice provided for development of gB-binding and HCMV neutralizing assays received from Dr. Barbara Kropff (Univ. Erlangen) and Prof. Christian Sinzger (Univ. Ulm). Prof. Arnold Ganser and Dr. Michael Stadler (MHH, Clinic of Hematology, Hemostasis, Oncology and Stem Cell Transplantation) kindly provided under informed consent patient material as a control reference for gB staining. Prof. Andre Bleich and Dr. Dirk Wedekind (MHH, Animal Facility) assisted with animal study protocols and mice husbandry. We thank Martin Fricke (MHH, Department of Clinical Immunology and Rheumatology) for helping with autoantibody staining. We thank Dr. Kristina Howard (FDA) for important suggestions regarding the experimental set-up for immune-depleting B cells from humanized mice. The authors thank other members of the Regenerative Immune Therapies Applied Laboratory, in particular Laura Gerasch and Benjamin Ostermann, for their valuable technical contributions.

## Author Contributions

**Conceptualization:** Martin Messerle, Florian Klein, Renata Stripecke.

**Data curation:** Sebastian J. Theobald, Christoph Kreer.

**Formal analysis:** Sebastian J. Theobald, Christoph Kreer, Sahamoddin Khailaie, Frank Klawonn, Michael Meyer-Hermann, Florian Klein, Renata Stripecke.

**Funding acquisition:** Martin Messerle, Florian Klein, Renata Stripecke.

**Investigation:** Sebastian J. Theobald, Christoph Kreer, Constanca Figueiredo, Johannes Koenig, Henning Olbrich, Andreas Schneider, Valery Volk, Simon Danisch, Lutz Gieselmann, Meryem Seda Ercanoglu, Torsten Witte.

**Methodology:** Sebastian J. Theobald, Christoph Kreer, Agnes Bonifacius, Britta Eiz-Vesper, Constanca Figueiredo, Michael Mach, Marija Backovic, Matthias Ballmaier, Torsten Witte.

**Project administration:** Renata Stripecke.

**Resources:** Constantin von Kaisenberg, Florian Klein, Renata Stripecke.

**Software:** Christoph Kreer, Sahamoddin Khailaie, Michael Meyer-Hermann.

**Supervision:** Michael Meyer-Hermann, Florian Klein, Renata Stripecke.

**Validation:** Sebastian J. Theobald, Christoph Kreer.

**Visualization:** Sebastian J. Theobald, Christoph Kreer, Sahamoddin Khailaie, Michael Meyer-Hermann, Renata Stripecke.

**Writing – original draft preparation:** Sebastian J. Theobald, Christoph Kreer, Renata Stripecke.

**Writing – review & editing:** Sebastian J. Theobald, Christoph Kreer, Sahamoddin Khailaie, Martin Messerle, Frank Klawonn, Michael Meyer-Hermann, Florian Klein, Renata Stripecke.

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
