## [Decision Letter · Decision Letter 0]

10 Dec 2019

Dear Prof Stripecke,

Thank you very much for submitting your manuscript "Repertoire characterization of functional human IgGs directly cloned from humanized mice vaccinated with dendritic cells and protected against HCMV" (PPATHOGENS-D-19-01985) for review by PLOS Pathogens. Your manuscript was fully evaluated at the editorial level and by independent peer reviewers. The reviewers appreciated the attention to an important problem, but raised some substantial concerns about the manuscript as it currently stands. These issues must be addressed before we would be willing to consider a revised version of your study. We cannot, of course, promise publication at that time.

We therefore ask you to modify the manuscript according to the review recommendations before we can consider your manuscript for acceptance. Your revisions should address the specific points made by each reviewer.

(1) A letter containing a detailed list of your responses to the review comments and a description of the changes you have made in the manuscript. Please note while forming your response, if your article is accepted, you may have the opportunity to make the peer review history publicly available. The record will include editor decision letters (with reviews) and your responses to reviewer comments. If eligible, we will contact you to opt in or out.

(2) Two versions of the manuscript: one with either highlights or tracked changes denoting where the text has been changed; the other a clean version (uploaded as the manuscript file).

Additionally, to enhance the reproducibility of your results, PLOS recommends that you deposit your laboratory protocols in protocols.io, where a protocol can be assigned its own identifier (DOI) such that it can be cited independently in the future. For instructions see http://journals.plos.org/plospathogens/s/submission-guidelines#loc-materials-and-methods

We hope to receive your revised manuscript within 60 days. If you anticipate any delay in its return, we ask that you let us know the expected resubmission date by replying to this email. Revised manuscripts received beyond 60 days may require evaluation and peer review similar to that applied to newly submitted manuscripts.

[LINK]

Sincerely,

Christopher M. Snyder, Ph.D.

Guest Editor

PLOS Pathogens

Blossom Damania

Section Editor

PLOS Pathogens

Kasturi Haldar

Editor-in-Chief

PLOS Pathogens

orcid.org/0000-0001-5065-158X

Grant McFadden

Editor-in-Chief

PLOS Pathogens

orcid.org/0000-0002-2556-3526

All 3 reviewers found this work to be interesting and significant. However, all 3 felt that additional experiments and discussion are necessary. Please respond to each point and particularly address the comments from each reviewer that some in vivo test of protection mediated by the antibodies is necessary.

Reviewer's Responses to Questions

**Part I - Summary**

Reviewer #1: The authors demonstrate that dendritic cells that are induced by GM-CSF and IFN-alpha from monocytes (iDCs) and transgenic for the gB envelope protein (iDC-gB) of the human cytomegalovirus (HCMV) protect humanized mice from challenge by HCMV infection. This protection is associated with elevated CD4+ T cell numbers in the bone marrow and IgA+ or IgG+ B cells in liver or spleen, respectively. They then characterized IgG of splenic B cells after iDC-gB immunization with and without HCMV infection and cloned nine antibodies from the different experimental conditions. While the antibody derived from HCMV infection alone was not able to neutralize HCMV, a subset of antibodies from iDC-gB immunized mice with or without HCMV infection were neutralizing. Their neutralizing IC50 was comparable to polyclonal human serum, but 100fold lower than for a monoclonal gB specific antibody (SM5-1). These are encouraging results suggesting that protective antibody responses against HCMV infection can be induced in humanized mice.

While the presented results are in line with several recent studies that suggest IgG antibody production after immunization in humanized mice, their role in the observed immune control of HCMV should be better characterized. The study suggests that protective antibody responses can be induced. However, this suggestion should be proven either by B cell depletion or adoptive antibody transfer.

Reviewer #2: A strategy to vaccinate against HCMV is a top priority. A number of vaccines have been trialled or in trial with varying degrees of success. This study builds on a focus of the lab looking to use DCs engineered to express HCMV antigens and cytokines to drive immune responses. they have already published the approach for pp65 where they demonstrated that T cells against pp65 were not effective. Thus the novelty and significance here is not the approach but the outcome - gB immune responses contribute to control of HCMV and that B cells could be rapidly cloned out to identify antibody sequences. In line with this the authors speculate that the control is humoral and provide evidence in support of that.

The studies in general look to be have been well done. There are a lot of data presented and sometimes it is hard to pick out the key data as the presentation could have been better (it may be my pdf viewer but some of the graphs were blurred making it hard to visualise some of the data points).

One aspect was that it was somewhat surprising that no discussion in light of the multiple attempts to use gB as a vaccine in humans and other animal models (and downstream attempts to understand how those vaccines worked) have been published but little mention is made of them. I am thinking of work from Adler, Bernstein, Schleiss, Diamond, McAvoy, Griffiths, Reeves, Permar etc that all have investigated the basis of immune responses against gB. Parallels could eaily be drawn.

Reviewer #3: Theobald et al. present a manuscript on engineered dendritic cells expressing granulocyte-macrophage colony-stimulating factor, interferon-alpha and HCMV-gB (iDCgB) as a model for immunotherapy to promote antibody responses against human Cytomegalovirus. This bio-engineered cell system enable effective anti-HCMV protection and isolation of human virus-neutralizing monoclonal antibodies. This is a topic of high clinical relevance. Although pharmacologic treatment has improved over the last decades, HCMV is still a problem with high mortality among high risk stem cell transplant recipients.

**Part II – Major Issues: Key Experiments Required for Acceptance**

Reviewer #1: 1. In order to demonstrate protection by the elicited humoral immune response, gain- or loss-of-function experiments should be performed. The authors could either deplete B cells with anti-CD20 antibodies to possibly abrogate protection against HCMV infection, or adoptively transfer one of their isolated neutralizing antibodies into HCMV infected humanized mice to demonstrate that these antibodies mediate immune control in vivo.

2. The authors demonstrate that their iDC-gB treatment increases CD4+ T cell numbers and especially in the bone marrow these correlate with protection from HCMV challenge, but they never attempt to demonstrate HCMV or even gB specificity of these T cell populations. Do these CD4+ T cells produce cytokines in response to re-stimulation with HCMV antigens? Is especially IL-21 production observed? Are follicular helper CD4+ T cells increased in number after iDC-gB treatment?

3. In order to judge the potency of the described method to induce IgG responses some additional characteristics of the cloned antibodies should be revealed. Even so it is mentioned in the abstract that only low to moderate levels of somatic hypermutation were observed, no concrete numbers seem to be given in the manuscript. Especially the number of somatic hypermutations in the isolated nine antibodies should be reported. Do these correlate with gB binding affinity or neutralization capacity?

Reviewer #2: 1. One major point that needs clarification is the measure of control. My understanding is that the mice are injected with humanised CD34+ cells, vaccinated with the DCs/gB, and then challenged with MRC5 cells infected with HCMV. I presume the premise is that the HCMV will then go latent in the CD34+ cells and thus the authors measure reactivation (similar to the model used by Nelson and colleagues). My question is the establishment of latency the same in all mice? Is this measured? Hypothetically, the MRC5 cell infection is controlled better in the vaccinated mice? If this led to less latency then it would read out as less reactivation. so the control is of the challenge not reactivation? could the authors clarify? this is related to point 3 below where it is important to know if the antibodies are controlling initial viraemia or controlling reactivation? It is possible that in different settings different immune responses play a role? e.g. reactivation in DCs would be considered to be more cell associated and more resistant to antibodies whereas MRC5s likely make cell free virus where neutralising abs would be more effective? (the authors state work from Stanton's lab so are aware of this aspect of CMV biology)

2. Leading on from that is it possible to challenge the mice with MCMV? Presumably the gB immune response will work against MCMV? this would allow the authors to directly analyse primary infection

3. Finally, the authors show evidence of neutralisation. A report in Science suggested the control of CMV reactivation by antibodies was due to non-neutralising functions of antibodies. The authors imply the neutralising response is not uniform from all mice so is it possible to correlate neutralising capacity with viraemia/reactivation in the mice? Alternatively can the antibodies be used to provide passive immunity in the model?

Reviewer #3: The authors nicely show the generation of specific antibodies. However, testing of these antibodies is limited to in vitro experiments. In vivo analysis would be important to support the conclusion and relevance stated by the authors.

The reader would be interested in experiments combining induction of a T cell response (eg against pp65) together with the iDCgB.

What is the translational perspective for the approach? Currently hyperimmunoglobulin agents against HCMV have been shown to have low or absent efficacy. They are not recommended in evidence based recommendations. However, these clinically available agents are of poor quality and specificity. The authors should provide in vivo evidence, that their identified antibodies are superior to these agents.

The authors correctly stated, that antibodies and T-cell responses are responsible for protective immunity against HCMV. Do iDCgB induce specific T helper cell responses? Using these DC for vaccination should induce also at least a CD4+ specific response to maintain a sustained protection in vivo. Presence of antibodies alone will hardly protect patients, since immunoglobulin treatment alone cannot prevent HCMV reactivation.

**Part III – Minor Issues: Editorial and Data Presentation Modifications**

Reviewer #1: 1. In the title the authors state that they analyzed functional human IgGs. Are there non-functional IgGs? What function do they mean? I would rather specify that they characterized HCMV gB specific human IgGs.

Reviewer #2: A minor issue is whether all the CD34+ cells were from HCMV seronegative donors? Can this be confirmed in the manuscript as of course that could impact on the response observed?

also were T cell responses against gB detected? I presume they were made. Have the authors ruled out T cell responses as contributing to control?

Reviewer #3: (No Response)

PLOS authors have the option to publish the peer review history of their article (what does this mean?). If published, this will include your full peer review and any attached files.

Reviewer #1: Yes: Christian Münz

Reviewer #2: No

Reviewer #3: No

---

## [Decision Letter · Decision Letter 1]

28 Mar 2020

Dear Prof Stripecke,

Thank you very much for submitting your manuscript "Repertoire characterization and validation of gB-specific human IgGs directly cloned from humanized mice vaccinated with dendritic cells and protected against HCMV" for consideration at PLOS Pathogens. As with all papers reviewed by the journal, your manuscript was reviewed by members of the editorial board and by several independent reviewers. The reviewers appreciated the attention to an important topic. Based on the reviews, we are likely to accept this manuscript for publication, providing that you modify the manuscript according to the review recommendations.

The reviewers have found this study to be significant and important and are supportive of its publication. Reviewer 2 raises an important point for clarification in the text about the level of latency between vaccinated and unvaccinated mice prior to reactivation studies. Reviewer 1 raises some concerns about the statistical significance of the new data shown in Figures 3e-f and Figure 5m-p. Complete replicate experiments would solidify this portion of the study. However, in lieu of performing complete replicates, the editors suggest that the authors may wish to soften their conclusions throughout (i.e. terms like "integral participants"–line 294-296). Rather, the authors might expand which aspects of the work are conclusive and which would fall under a proof-of-concept (line 469) category, citing the p-values of the data, and/or acknowledging the limitations of this part of the study and remaining unknowns about the role of antibodies in this model.

Sincerely,

Christopher M. Snyder, Ph.D.

Guest Editor

PLOS Pathogens

Blossom Damania

Section Editor

PLOS Pathogens

Kasturi Haldar

Editor-in-Chief

PLOS Pathogens

orcid.org/0000-0001-5065-158X

Michael Malim

Editor-in-Chief

PLOS Pathogens

orcid.org/0000-0002-7699-2064

The reviewers have found this study to be significant and important and are supportive of its publication. Reviewer 2 raises an important point for clarification in the text about the level of latency between vaccinated and unvaccinated mice prior to reactivation studies. Reviewer 1 raises some concerns about the statistical significance of the new data shown in Figures 3e-f and Figure 5m-p. Complete replicate experiments would solidify this portion of the study. However, in lieu of performing complete replicates, the editors suggest that the authors may wish to soften their conclusions throughout (i.e. terms like "integral participants"–line 294-296). Rather, the authors might expand which aspects of the work are conclusive and which would fall under a proof-of-concept (line 469) category, citing the p-values of the data, and/or acknowledging the limitations of this part of the study and remaining unknowns about the role of antibodies in this model.

Reviewer Comments (if any, and for reference):

Reviewer's Responses to Questions

**Part I - Summary**

Reviewer #1: The authors demonstrate that dendritic cells that are induced by GM-CSF and IFN-alpha from monocytes (iDCs) and transgenic for the gB envelope protein (iDC-gB) of the human cytomegalovirus (HCMV) protect humanized mice from challenge by HCMV infection. This protection is associated with elevated CD4+ T cell numbers in the bone marrow and IgA+ or IgG+ B cells in liver or spleen, respectively. They then characterized IgG of splenic B cells after iDC-gB immunization with and without HCMV infection and cloned nine antibodies from the different experimental conditions. While the antibody derived from HCMV infection alone was not able to neutralize HCMV, a subset of antibodies from iDC-gB immunized mice with or without HCMV infection were neutralizing. Their neutralizing IC50 was comparable to polyclonal human serum, but 100fold lower than for a monoclonal gB specific antibody (SM5-1). These are encouraging results suggesting that protective antibody responses against HCMV infection can be induced in humanized mice.

Reviewer #2: This remains an interesting study and for the most part the additional experiments requested have been done and they contribute to importance of the study.

**Part II – Major Issues: Key Experiments Required for Acceptance**

Reviewer #1: In the revised manuscript the authors provide additional experimental data that address all three of my original concerns. Namely, they have performed B cell depletions, adoptive transfer of a combination of two of their isolated HCMV specific antibodies, and describe now better that the limited somatic hypermutation that they observe does not seem to have increased the neutralizing ability of the respective antibodies. Unfortunately, the new experiments (B cell depletion and adoptive antibody transfer) seem to have been performed only once with limited numbers of mice per experimental group. Therefore, statistical evaluation of the observed differences could not be performed. Both experiments should be at least repeated once and statistically evaluated.

Reviewer #2: The only remaining issue I have is regarding my question regarding the establishment of latency which, based on the authors response, may not have been made wholly clear.

What isn't clear to me when reading the manuscript was whether the original establishment of latency was measured in the animals who were treated with control or the vaccine prior to infection. It stands to reason that if latency establishment is impaired by an anti-HCMV immune response then when reactivation analyses are performed there will be an effect seen but it may not necessarily be a block to reactivation but more reflect a difference on the initial impact on the seeding of latency in the model.

I appreciate it may not be easy to go back and assess latency if no tissue is available but I think it is worth noting in the discussion this limitation that the original establishment of latency may be impaired by the vaccine treatment unless there are data on latent load I have missed.

**Part III – Minor Issues: Editorial and Data Presentation Modifications**

Reviewer #1: None

Reviewer #2: no minor comments

PLOS authors have the option to publish the peer review history of their article (what does this mean?). If published, this will include your full peer review and any attached files.

Reviewer #1: Yes: Christian Münz

Reviewer #2: No
---

## [Editor Report · Decision Letter 2]

9 Apr 2020

Dear Prof Stripecke,

Thank you very much for re-submitting your manuscript "Repertoire characterization and validation of gB-specific human IgGs directly cloned from humanized mice vaccinated with dendritic cells and protected against HCMV" for consideration at PLOS Pathogens. 

I think there is still a mis-understanding about the clarification requested by Reviewer #2 in the prior review. This reviewer is referring to this prophylactic vaccination setting described in this study, and asking whether the iDC vaccinations conducted between weeks 6 and 10, affected the total HCMV latent loads established after the week 17 HCMV infection, but prior to the reactivation started at week 24. In other words, just before the G-CSF treatment in week 24, do all of the vaccinated and unvaccinated mice have equivalent amounts of viral DNA? Or is there less viral DNA in the vaccinated mice compared to the controls? If the vaccinated mice controlled the primary HCMV infection in week 17 better, it could result in reduced latent viral DNA by week 24, which could be the cause of the reduced luciferase signal after reactivation (e.g. in Figure 2d). Thus, the vaccination might be having its greatest effect on the primary infection and the establishment of latency after week 17, rather than on the reactivation from latency after week 24.

Sincerely,

Christopher M. Snyder, Ph.D.

Guest Editor

PLOS Pathogens

Blossom Damania

Section Editor

PLOS Pathogens

Kasturi Haldar

Editor-in-Chief

PLOS Pathogens

orcid.org/0000-0001-5065-158X

Michael Malim

Editor-in-Chief

PLOS Pathogens

orcid.org/0000-0002-7699-2064

I think there is still a mis-understanding about the clarification requested by Reviewer #2. This reviewer is referring to this prophylactic vaccination setting described in this study, and asking whether the iDC vaccinations conducted between weeks 6 and 10, affect the total HCMV latent loads established after the week 17 HCMV infection, but prior to the reactivation started at week 24. In other words, just before the G-CSF treatment in week 24, do all of the vaccinated and unvaccinated mice have equivalent amounts of viral DNA? For example, if the vaccinated mice controlled the primary HCMV infection in week 17 better, it could result in reduced latent viral DNA by week 24, which could then lead to reduced luciferase signal after reactivation (e.g. in Figure 2d). Thus, the vaccination might be having its greatest effect on the primary infection and the establishment of latency after week 17, rather than on the reactivation from latency after week 24.
---

## [Editor Report · Decision Letter 3]

18 Apr 2020

Dear Prof Stripecke,

We are pleased to inform you that your manuscript 'Repertoire characterization and validation of gB-specific human IgGs directly cloned from humanized mice vaccinated with dendritic cells and protected against HCMV' has been provisionally accepted for publication in PLOS Pathogens.

Best regards,

Christopher M. Snyder, Ph.D.

Guest Editor

PLOS Pathogens

Blossom Damania

Section Editor

PLOS Pathogens

Kasturi Haldar

Editor-in-Chief

PLOS Pathogens

orcid.org/0000-0001-5065-158X

Michael Malim

Editor-in-Chief

PLOS Pathogens

orcid.org/0000-0002-7699-2064

Thank you for the revisions and clarifications.
---

## [Editor Report · Acceptance letter]

4 Jun 2020

Dear Prof. Stripecke,

We are delighted to inform you that your manuscript, "Repertoire characterization and validation of gB-specific human IgGs directly cloned from humanized mice vaccinated with dendritic cells and protected against HCMV," has been formally accepted for publication in PLOS Pathogens.

Best regards,

Kasturi Haldar

Editor-in-Chief

PLOS Pathogens

orcid.org/0000-0001-5065-158X

Michael Malim

Editor-in-Chief

PLOS Pathogens

orcid.org/0000-0002-7699-2064